# Antibody-mediated NK cell activation as a correlate of immunity against influenza infection

Carolyn M. Boudreau[1,2], John S. Burke IV[1], Ashraf S. Yousif[1], Maya Sangesland[1,2], Sandra Jastrzebski[3], Chris Verschoor [4], George Kuchel [3], Daniel Lingwood [1], Harry Kleanthous [5], Iris De Bruijn[6], Victoria Landolfi[7], Saranya Sridhar [7,8] ✉ & Galit Alter[1,8] ✉

Antibodies play a critical role in protection against influenza; yet titers and viral neutralization represent incomplete correlates of immunity. Instead, the ability of antibodies to leverage the antiviral power of the innate immune system has been implicated in protection from and clearance of influenza infection. Here, post-hoc analysis of the humoral immune response to influenza is comprehensively profiled in a cohort of vaccinated older adults (65 + ) monitored for influenza infection during the 2012/2013 season in the United States (NCT: 01427309). While robust humoral immune responses arose against the vaccine and circulating strains, influenza-specific antibody effector profiles differed in individuals that later became infected with influenza, who are deficient in NK cell activating antibodies to both hemagglutinin and neuraminidase, compared to individuals who remained uninfected. Furthermore, NK cell activation was strongly associated with the NK cell senescence marker CD57, arguing for the need for selective induction of influenza-specific afucosylated NK activating antibodies in older adults to achieve protection. High dose vaccination, currently used for older adults, was insufficient to generate this NK cell-activating humoral response. Next generation vaccines able to selectively bolster NK cell activating antibodies may be required to achieve protection in the setting of progressively senescent NK cells.

Over the past decades, seasonal influenza has caused increasing numbers of deaths in the United States, explained in part by the aging population[1]. Older adults suffer from a disproportionate burden of severe disease and hospitalization following influenza infection, even in the setting of broad vaccine campaigns[2]. While seasonal vaccination has resulted in a 40% reduction in hospitalizations in this vulnerable population[3], the current vaccines are estimated to provide between 10 and 60% protection annually[4]. During the 2012-2013 influenza season, the vaccine achieved a respectable 52% protection against medically-attended acute respiratory disease across all age groups, albeit with lower efficacy in people 65 and older[5]. Correlate analyses observed that hemagglutination inhibition (HAI) accounted for only ~60% of the observed protection[6–8], pointing to the importance of additional, non-HAI vaccine mechanisms as additional correlates of immunity.

[1]Ragon Institute of MGH, MIT, and Harvard, Cambridge, MA 02129, USA. [2]PhD Program in Virology, Division of Medical Sciences, Harvard University, Boston, MA 02115, USA. [3]Center on Aging, UCONN Health Center, Farmington, CT 06030, USA. [4]Department of Pathology and Molecular Medicine, Faculty of Health Sciences, McMaster University, Hamilton, ON L8S 4L8, Canada. [5]SK Bioscience, Cambridge, MA 02141, USA. [6]Sanofi-Pasteur, Inc., Marcy-l'Étoile, France. [7]Sanofi-Pasteur, Inc., Cambridge, MA 02129, USA. [8]These authors contributed equally: Saranya Sridhar, Galit Alter. ✉e-mail: saranya.sridhar@sanofi.com; ragonsystemserology@mgh.harvard.edu

Moreover, given the mismatch in the H3N2 vaccine antigen and the circulating strain during the 2012–2013 season[9] that resulted in compromised neutralization, it was highly likely that alternate, non-neutralizing, vaccine-induced immune mechanisms played a critical role as contributors to protection. Thus, the 2012-2013 season offered a unique opportunity to define correlates of immunity against influenza beyond neutralization.

The identification of broadly reactive monoclonal antibodies, such as CR6261[10] and CR9114[11], that provide robust in vivo protection[12,13] even in the absence of neutralization have raised the possibility that additional humoral mechanisms beyond neutralization may be critical for immunity against influenza[14–16]. Several antibody-effector functions have been implicated in protection from influenza infection[16], including antibody-mediated macrophage phagocytosis[17,18], neutrophil activation[17], antibody-dependent cellular cytotoxicity (ADCC)[19] by NK cells, monocytes, and neutrophils, and complement-specific lysis[18,20–22]. Hemagglutinin (HA)-specific functional antibodies capable of inducing ADCC arise early following infection and exhibit more breadth of binding compared to neutralizing antibodies[23], and have been shown to decrease symptom severity and increase viral clearance[24]. Furthermore, in older adults, increased ADCC activity correlated with strong HAI responses following vaccination[25], although HAI antibodies have been proposed to compete with ADCC antibodies for access to antigenic epitopes[26]. However, whether ADCC alone is critical for protection, or whether ADCC antibodies work in tandem with other antibody-effector functions to provide maximal protection from infection, remains incompletely understood.

To objectively begin to define the functional humoral correlates of immunity against influenza, we deeply profiled the functional humoral immune response to influenza at 28 days post immunization in a unique cohort of vaccinated older adults sampled and monitored during the 2012–2013 season[8]. Influenza-specific antibody-effector profiles at peak immunogenicity were markedly different between individuals that later became infected with influenza. Individuals who became infected had responses marked by a highly selective deficit of NK cell activating antibodies to both the hemagglutinin (HA) and neuraminidase (NA) antigens of H3N2 influenza, compared to the individuals who remained uninfected. Linked to the observed progressive loss of ADCC activity in older adults, the data presented here provide strong evidence for a preferential and critical role for Fc-afucosylated NK cell-recruiting antibodies in protection against influenza in the elderly. Thus, next-generation vaccines able to selectively bolster NK cell-activating antibodies may be required to achieve protection in the setting of progressively senescent NK cells.

## Results

### Protected vaccinees induce coordinated humoral immune responses

Older adults (≥65 years old) were enrolled in the FIM12 efficacy trial[8,27] and immunized with either a regular or high-dose trivalent influenza vaccine in October of 2012. Serum samples were collected 28 days post vaccination. No baseline samples were collected; therefore, all samples reflect both vaccine-induced and previously occurring immunity. All vaccinees were tracked throughout the influenza season, with twice weekly surveillance calls during the peak of the influenza season (January-February 2013), followed by once weekly calls until the end of the season (30 April 2013). Any individuals with respiratory illness were tested for influenza by PCR, culture, or both. The majority of influenza-positive individuals were infected with influenza A subtype H3N2. No individuals with respiratory symptoms who tested negative for influenza were included in this study set. A small number ($n = 5$) of individuals who displayed respiratory symptoms yet did not meet clinical criteria for influenza-like illness but had influenza viral infections detected by PCR and/or culture were included. Samples from all

available cases and a set of controls, matched for age, sex, and vaccine dose, were included in this study[8] (Table S1).

In line with previous studies pointing to an incomplete association between HAI and protection[6–8], individuals who did not become infected during the study period (controls) trended toward higher post vaccination and pre-infection HAI titers towards the circulating H3N2 strain compared to the individuals who later became infected (cases), although this difference was not significant ($p = 0.075$, Fig. 1A). No differences were observed between HAI titers for other vaccine strains (Figure S1A, B). Additionally, no difference was observed in overall serum IgG titers against HA and NA from the circulating H3N2 strain (Fig. 1B). Importantly, differences in the magnitude of the antibody response were noted in overall vaccine-induced profiles across the vaccine antigen and circulating strain antigen (Figure S1C), which differed at three amino acids in the HA protein due to egg adaptation of both the regular and high-dose vaccine during this season[9,28], responses were highly correlated across H3N2 vaccine and circulating antigens (Figure S1D). These data suggested that alternate humoral qualities, beyond HAI and HA-specific binding titers, are likely associated with protection.

To gain a deeper sense of additional antibody correlates of immunity, systems serology[29] was utilized to measure antigen-specific antibody-dependent innate immune functions (Antibody-dependent cellular phagocytosis, ADCP; Antibody-dependent neutrophil phagocytosis, ADNP; Antibody-dependent complement deposition, ADCD; Antibody-dependent NK cell activation, ADNKA CD107a, IFN-γ, and MIP-1β; and Antibody-dependent DC phagocytosis, ADDCP CD83, and HLA-DR expression), antigen-specific antibody isotype titers, and antigen-specific Fcγ-receptor (FCGR) binding profiles at peak immunogenicity. All samples were profiled under blinded conditions, and researchers were unblinded for analysis once all data was collected. Serum samples from pre-vaccination timepoints were unavailable, thus antibody profiles described here reflect both vaccine-induced and naturally occurring humoral immune responses. Antibody Fc profiling across the cases and controls highlighted significant heterogeneity in vaccine-induced H3- and N2- specific Fc profiles across all vaccine recipients (Fig. 1C), with no two individuals possessing an identical HA- or NA-specific Fc profile.

While no single feature was associated with the risk of infection following multiple hypothesis correction, correlational analysis of all H3-specific Fc-effector functions highlighted distinctions between cases and controls (Fig. 1D). Specifically, enhanced coordination was observed in HA-specific antibody functions in the controls. Conversely, increased negatively correlated relationships were observed in the humoral profiles generated in plasma from cases. Cases were marked by the preservation of positive correlations across opsonophagocytic (ADCP, ADNP), complement-fixing (ADCD), and DC activating (ADDCP, HLA-DR, CD83, CD86) antibody functions, but these functions were largely inversely correlated with antibody-mediated NK cell functional antibody responses. These data point to the potential importance of the inclusion of NK cell functions in the protective polyfunctional antibody response to influenza, highlighting a possible role for antibody-mediated NK cell recruitment in protective immunity.

### Antibody-mediated NK cell activating antibody profiles separate cases from controls

The observed differences in antibody Fc effector function profiles across cases and controls, as highlighted by the distinct correlation patterns, suggested multivariate, rather than univariate, differences linked to protection from infection. A Least Absolute Shrinkage and Selection Operator (LASSO)-Elastic Net algorithm[30,31] was designed to select a minimal set of antibody features that accounted for the greatest variation in antibody profiles across the cases and controls. Features included in this analysis were Fc-mediated innate immune functions by HA- and NA-specific antibodies, FcR binding levels against

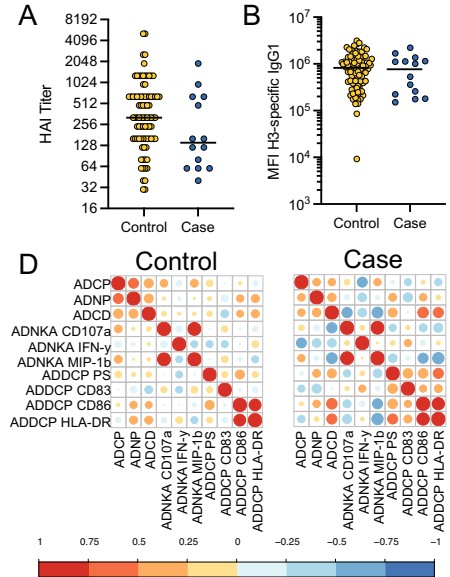

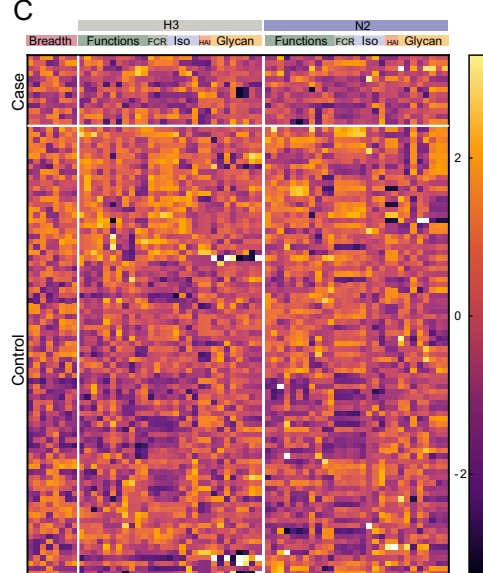

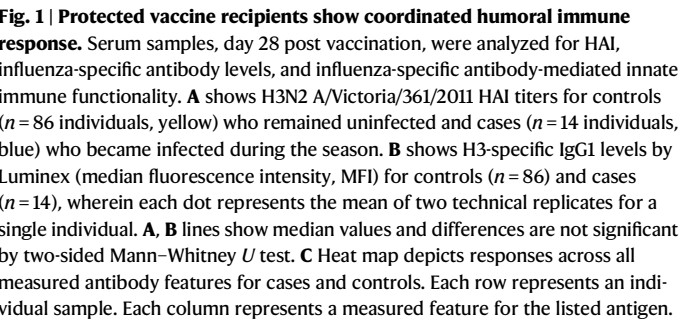

**Fig. 1 | Protected vaccine recipients show coordinated humoral immune response.** Serum samples, day 28 post vaccination, were analyzed for HAI, influenza-specific antibody levels, and influenza-specific antibody-mediated innate immune functionality. **A** shows H3N2 A/Victoria/361/2011 HAI titers for controls (*n* = 86 individuals, yellow) who remained uninfected and cases (*n* = 14 individuals, blue) who became infected during the season. **B** shows H3-specific IgG1 levels by Luminex (median fluorescence intensity, MFI) for controls (*n* = 86) and cases (*n* = 14), wherein each dot represents the mean of two technical replicates for a single individual. **A**, **B** lines show median values and differences are not significant by two-sided Mann−Whitney *U* test. **C** Heat map depicts responses across all measured antibody features for cases and controls. Each row represents an individual sample. Each column represents a measured feature for the listed antigen.

Breadth features reflect antigen-specific antibody isotype and FcR binding across all tested strains (Table S3). Values for all measurements were *Z* score normalized with *Z* score values depicted on the color map. Rows were manually clustered by infection status. ADCP antibody-dependent cellular phagocytosis, ADNP antibody-dependent neutrophil phagocytosis, ADCD antibody-dependent complement deposition, ADNKA antibody-dependent NK cell activation, ADDCP antibody-dependent Dendritic Cell phagocytosis. **D** Correlation matrices show Spearman R correlations between H3-specific antibody-dependent functions for controls and cases. The size of the circles and the color of the circles represent the strength of the correlation, with red for positive and blue for negative correlations. Source data are provided as a Source Data file.

all tested antibody specificities, antibody isotype levels against all tested antibody specificities, HAI, and NAI. This allowed identification of antibody profiles at peak vaccine-induced immunogenicity prior to exposure that were selectively and uniquely enriched in individuals that either ultimately became infected as compared to those who remained uninfected over the follow-up period. A partial least square discriminant analysis was then used to visualize whether the minimal features could discriminate the groups. Separation was observed in Fc profiles across cases and controls (Fig. 2A, *p* < 0.01 when compared to both permuted label and random size-matched feature sets, Figure S2A), with the groups separating largely across the *x* axis, or latent variable 1 (LV1). The LV1 score, which is a composite variable composed of multiple humoral features collapsed into one numeric representation, significantly differed across the cases and controls (Fig. 2B). This indicates that the LV1 variable is a predictive correlate of protection.

To understand the features that differed across the groups and begin to untangle their biological importance, the specific humoral measurements used to compose LV1 were ranked in order of their importance in discriminating groups. Four component features of LV1 were enriched in the controls and five features were enriched in the cases (Fig. 2C). IgG3, IgA1, and IgM antibodies directed to both HA and NA antigens were linked to non-protective immunity, suggesting that individuals who ultimately developed influenza symptoms mounted an immune response marked by the generation of de novo (IgM/IgG3) and mucosal (IgA1) humoral responses, which targeted several viral strains including H1N1, H3N2, and B viruses. The IgA1 and IgM signatures were linked to specific HA subtypes (Influenza B and H1N1, respectively), targeting strains that were not in circulation throughout the study[32]. Because the cases in this study were primarily infected with H3N2, these data suggest that de novo response to non-H3 vaccine strains may result in susceptibility to infection, potentially arising from

skewed immunodominance. To determine whether expanded H1N1 responses were related to previous imprinting, given the age of our participants, H1- and N1-specific antibody profiles across cases and controls were directly compared (Figure S2B). While none of the differences between cases and controls reached statistical significance, a trend towards higher H1-specific immunity was observed in cases across all tested strains, pointing towards a potentially skewed humoral immune response among cases compared to controls independently of immune imprinting.

The four features enriched in the controls all pointed to a single, specific functional signature. Rather than overall quantitative increases in antibody levels, protected individuals experienced a selective increase in NK cell-activating antibodies directed to both HA and NA, including higher levels of FCGR3A binding and NK cell-activating antibodies. Moreover, deeper analysis of the correlates of the LASSO-selected features highlighted the presence of a large NK cell cluster associated with FCGR3A-binding antibody titers that were enriched in controls, indicating that a broad FCGR3A-binding response was selectively expanded in controls (Figure S2C). Additionally, a broad polyclonal IgG1 response, reactive across HA strains, including binding to both the vaccine subtypes, the infecting subtype, and to heterologous viruses was observed among all study participants (Figure S2D).

Emerging data point to a critical role for both conserved HA-stalk and neuraminidase (NA)-specific immunity in protection against influenza[33–35]. Thus, we also examined any univariate differences in stalk or NA-specific responses across the cases and controls. No differences were observed in stalk-specific antibody profiles across the groups, with only a slight enrichment of FCGR2A-binding antibodies in cases (Figure S3A−H). In contrast, a trend towards higher levels of antibodies against N2 A/Hong Kong/4801/2014, N2 A/Texas/50/2012, and H3N2 NAI were observed among the controls, although no

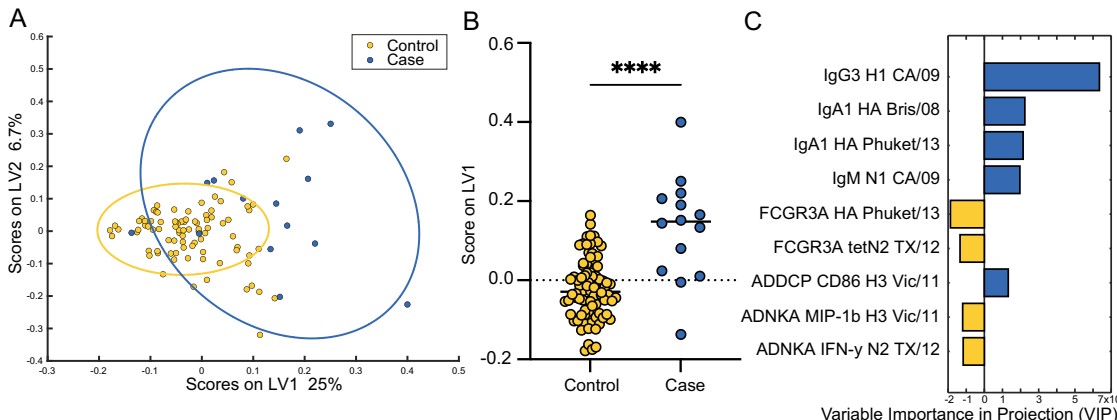

**Fig. 2 | Antibody-dependent NK cell activation is a predictor of protection from influenza infection. A** Partial least squares discriminant analysis separating cases (individuals who became infected; $n = 14$, blue) from controls (individuals who remained uninfected; $n = 86$, yellow) using a robust minimal set of antibody features that contribute to the disease outcome (LASSO-Elastic Net). Latent variables (LVs) describe combinations of humoral features, and scores across LVs indicate the percentage of separation described by that axis. Model is significant ($p < 0.01$) compared to the accuracy of permuted label and random size-matched models (Mann–Whitney $U$ test). **B** shows scores for individuals across LV1 with bars at median values for each group (cases $n = 14$; controls $n = 86$). Each dot represents an individual. Significance tested by two-sided Mann-Whitney $U$ test, ****$p < 0.0001$. **C** Variable importance in the projection plot indicates the relative contribution of humoral features to the model depicted in **A**. ADDCP antibody-dependent Dendritic Cell phagocytosis, ADNKA antibody-dependent NK cell activation. See also Figure S2. Source data are provided as a Source Data file.

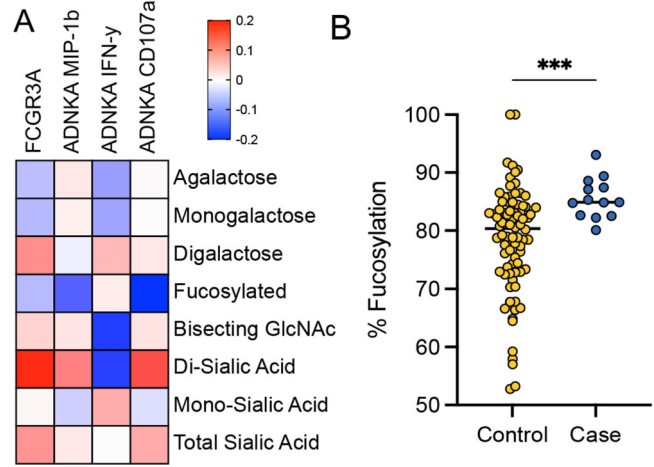

**Fig. 3 | Antibody Fc glycosylation linked to NK cell activation reflects influenza outcome.** Heat map **A** shows Spearman R correlations between measures of H3 WT-specific NK cell activation (ADNKA) and FCGR3A binding and H3 WT-specific antibody Fc glycoforms. Dot plot **B** shows levels of Fc fucosylation of H3 WT-specific antibodies in controls ($n = 86$, yellow) and cases ($n = 14$, blue) with lines at median values for each group. Significance tested by two-sided Mann–Whitney $U$ test, ***$p = 0.0007$. See also Figure S4. Source data are provided as a Source Data file.

differences in N1 A/Belgium/145-MA/2009 were observed across the cases and controls (Figure S3I–L). Taken together, these results suggest that H3- and NA-specific antibodies with the capacity to engage FCGR3A and activate NK cells are associated with lower susceptibility to influenza following vaccination in older adults and mice.

Correlation analysis across the antibody subclasses and functions suggested that NK cell activation and FCGR binding were primarily driven by the level of influenza-specific IgG1 levels (Figure S4A). Fc-glycosylation plays a critical role in modulating antibody affinity for Fc-receptors and driving effector function[36]. Specifically, changes in Fc-fucosylation have been linked to enhanced ADCC[37,38]. Thus, we next probed the correlation between H3-specific IgG glycosylation and NK cell activation (Fig. 3A). A negative correlation was observed between fucosylated antibodies and NK cell degranulation (CD107a

expression), a proxy for ADCC (Fig. 3A). Interestingly, bisecting-n-acetyl glucosamine (GlcNac) and sialylation (S) were also associated with enhanced NK cell degranulation but reduced NK cell cytokine secretion, potentially arguing for an association of particular afucosylated N-glycan structures with NK cell activation. However, to quantitatively explore the role of IgG glycosylation as a correlate of immunity, the levels of fucosylation were compared across cases and controls (Fig. 3B). Significantly increased levels of fucosylation were observed on HA-specific antibodies in cases compared to controls accompanied by increased di-sialylation and no change to bisecting GlcNAc levels (Figure S4B, C). Whether specific afucosylated glycan structures may differ more dramatically across cases and controls on particular epitope-specific antibody subpopulations remains unclear, but could contribute to enhanced NK cell activation and potential control of infection. Thus, while high-resolution glycopeptide analysis of HA-specific antibodies in the future is likely to identify precise N-glycan structures associated with NK cell activation, this data suggests that vaccination may be able to drive NK cell function by tuning antibody glycosylation towards afucosylation, enhancing NK cell activation and thereby conferring enhanced protection.

## High-dose vaccination enhances HAI but not the ability of antibodies to activate NK cells

The administration of higher doses of the influenza vaccine has been successfully utilized as a strategy to drive enhanced immunity in the aging population[8]. Thus, the impact of high-dose vaccination on specifically tuning the protective signature, NK cell responses, identified in this trial was explored. The impact of regular and high-dose vaccination on NK cell activating antibody levels and glycosylation was examined across cases and controls in each of the dose arms. While dosing improved HAI (Fig. 4A), it did not affect HA-specific antibody levels (Fig. 4B) and did not appear to drive an overall significant pattern of antibody Fc profiles (Fig. 4C). Moreover, closer inspection of NK cell functions and Fc-glycosylation revealed a trend, but not a significant improvement with the increased dose across both circulating HA (Fig. 4D–G) and NA antigen-specific antibodies (Figure S5) after multiple test correction. These data suggest that while high-dose vaccination may increase HAI significantly, dosing alone may be insufficient to fully enhance the level of NK cell-recruiting antibodies. Instead, next-generation strategies using distinct platforms

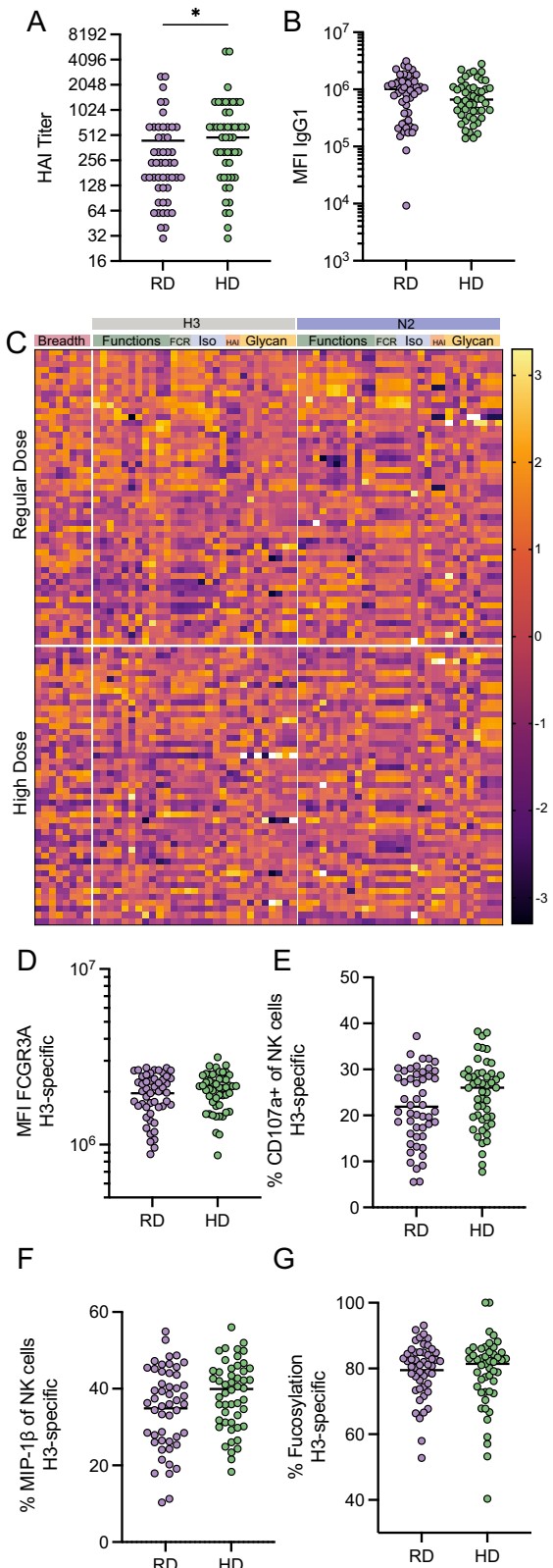

**Fig. 4 | High-dose vaccination enhances HAI but not antibody-mediated NK cell activation. A** shows H3N2 A/Victoria/361/2011 HAI titers for regular-dose vaccinees and high-dose vaccinees. **B** shows H3-specific IgG1 levels by Luminex (median fluorescence intensity, MFI) for RD (purple) and HD (green) recipients. **A**, **B** Each dot represents the mean of two technical replicates for a single individual. Bars represent medians and significance tested by two-sided Mann−Whitney U test, *$p < 0.05$. **C** Heat map depicts responses across all measured antibody features for RD and HD recipients. Each row represents an individual sample. Each column represents a measured feature. Values for all measurements were z score normalized. **D** shows H3-specific FCGR3A-binding levels for regular and high-dose vaccinees. **E** shows H3-specific NK cell CD107a expression for regular and high-dose vaccinees. **F** shows H3-specific NK cell MIP-1b expression for regular and high-dose vaccinees. **G** shows percentage of H3-specific antibodies that have fucosylated Fc glycans. **D−G** Each dot represents the mean of two technical replicates for a single individual. Bars represent medians and significance tested by Mann−Whitney U test, not significant. **A−G** regular dose vaccinees $n = 50$; high-dose vaccinees $n = 50$ individuals. See also Figure S5. Source data are provided as a Source Data file.

## NK cells in older adults are less responsive to antibody-mediated activation

Emerging data point to a gradual exhaustion of the NK cells with age[39] that may alter the level of NK cell activating antibodies required to mediate protection. Thus, we finally aimed to determine whether a simple biomarker could be defined to guide next generation vaccine development to induce sufficient levels of ADCC activity to drive protection. Thus, the ability of an ADCC-inducing antibody to activate NK cells was tested in PBMCs from a separate cohort of healthy younger (<40) or older (>65) adults not recently vaccinated for influenza (Fig. 5A). As expected[40,41], NK cells from older individuals expressed elevated levels of CD57, possibly a marker of senescence (Fig. 5B). However, limited differences were noted in FCGR3A (CD16) expression on NK cells across the ages (Fig. 5C). To determine whether age-related differences existed in ADCC activity, PBMCs were stimulated with an anti-FCGR3A (anti-CD16) activating antibody, a surrogate of activating ADCC. While NK cell activation was not directly linked to age (for CD107a Spearman's $R = -0.2347$, ns; for MIP-1b Spearman's $R = -0.0153$, ns), we observed a significant inverse correlation between CD57 expression and the degree of FCGR3A-mediated NK cell degranulation (CD107a), as well as an inverse correlation between CD57 and NK cell functional receptor expression including KIR and the C-type lectin, NKG2C levels (Fig. 5D−F, Figure S6). Thus, in line with our emerging appreciation between the disconnect in numerical and biologic age[42–44], NK cell senescence, marked by CD57 expression, was linked to defects in NK cell activation via FCGR3A. Conversely, no association with NK cell function or phenotype was observed with BMI or clinical frailty. These data argue that the degree of NK cell senescence, as measured by CD57 but not by age or frailty alone, is a marker of NK cell activatability, providing a biomarker to guide ADCC-inducing vaccine development for aging populations. Thus, enhanced CD57 levels may mark the degree of NK cell senescence and reflect the need for influenza vaccines to induce higher levels of NK cell activating antibodies in older adults to fully harness the NK antiviral activity that may be key to controlling and clearing the virus.

## Discussion

In a meta-analysis of 15 publications[45], vaccine efficacy increased by approximately 15% with high-dose vaccination, significantly reducing hospitalization in the > 65-year-old age group. While elevated dosing improved HAI and protection[8], this study found that dosing did not alter the level of antibodies able to drive NK cell degranulation following vaccination in this population, who already had exposure to influenza through vaccination and natural infection. Similarly, previous efforts to boost immunity with the use of an adjuvant also

and adjuvants may help improve the quality of vaccine-induced antibody functions, aimed at driving highly protective antibodies against influenza. However, the specific level of NK recruiting antibody required to drive protection in the elderly population remains incompletely understood.

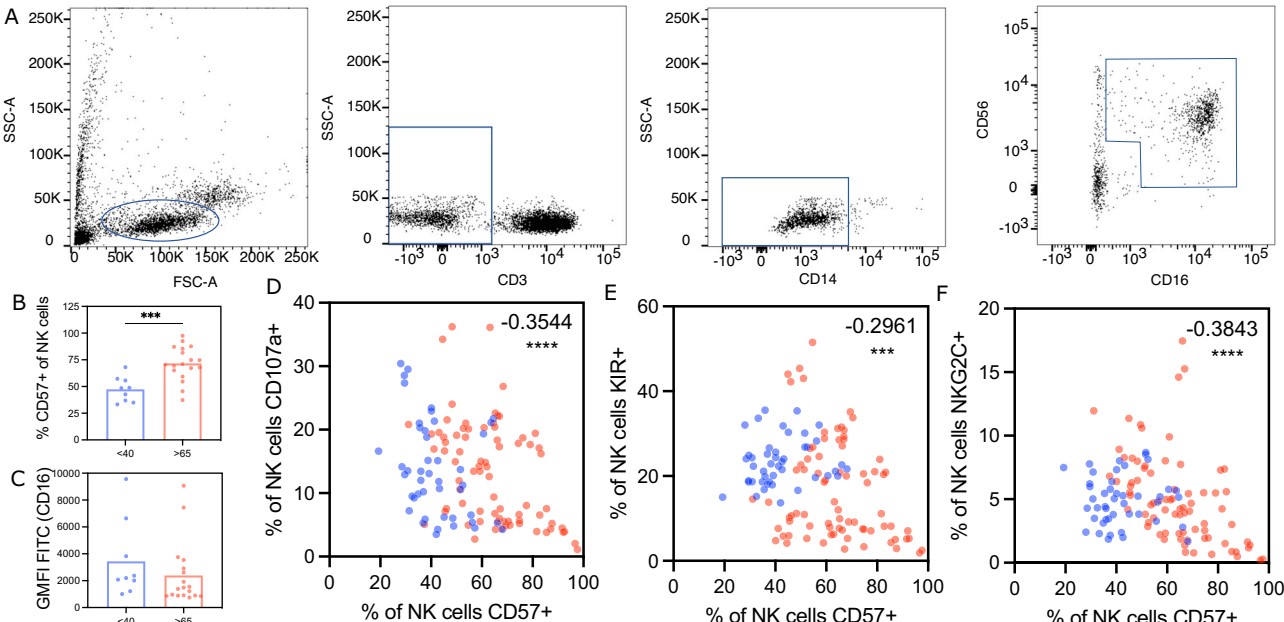

**Fig. 5 | CD57 as a biomarker for aging NK cells less responsive to FCGR3A-mediated stimulation.** Density plots **A** show representative gating for NK cells in Fig. 5. NK cells were defined by Size>Single Cells>Live>CDcd3−>CD14−>CD16+CD56+. Bar plots show the percentage of NK cells positive for CD57 (**B**) and CD16 (**C**) at baseline condition grouped by age. Dots show individual subjects (<40 n = 10 in blue, >65 $n$ = 19 in red) and bars show mean values for each group. GMFI geometric mean fluorescence intensity. Significance tested by two-sided $t$ test, ***$p$ = 0.0004. Correlation plots show correlations between CD57 and CD107a (**D**), KIR (**E**), or NKG2C (**F**) expression across CD16 stimulation conditions (0–10 µg/ml). Correlation measured by Spearman's R. ***$p$ = 0.0009, ****$p$ < 0.0001. See also Figure S6. Source data are provided as a Source Data file.

resulted in enhanced antibody titers and opsonophagocytic functions but led to poor induction of NK cell-activating antibodies[46]. However, more recent data suggest that distinct adjuvants have the capability of differentially harnessing antibody-effector functions, with the TLR7-agonist AS37, driving the highest levels of NK cell activating antibodies in a COVID-19 vaccine study[47]. Interestingly, because isotypes changed only marginally across distinct vaccine adjuvants[47], these data argue for adjuvant-mediated Fc-glycosylation changes as a key mechanism that may lead to distinct antibody-effector functions[48,49]. Specifically, here, lower levels of Fc-fucosylation were linked to both enhanced FCGR3A binding and NK cell activation, all of which were enriched in controls compared to cases. Given that dosing did not significantly alter Fc-fucosylation, additional strategies, including the selection of particular adjuvants or novel vaccine platforms may be key to enhancing protection in the aging population. However, whether durable changes in Fc-glycosylation can be induced via vaccination across the ages is unknown but could represent a unique opportunity to shape immunity to influenza and beyond[50,51]. Further studies will be required to understand these changes, neutralization beyond receptor blocking (HAI), the precise glycan structures required to induce NK cell activation, their durability, whether pre-existing antibody profiles influence the degree or quality of Fc-glycosylation, and how these may be induced preferentially via vaccination, which could not be fully explored in this study lacking baseline and longitudinal samples.

A unique observation in this study relates to the emergence of protective ADCC signals against both the influenza HA and NA proteins. While both HA and NA proteins are included in the seasonal inactivated influenza vaccine, only the concentration of HA is controlled at the time of vaccine production. Due at least in part to variability in NA content[52,53], typical seasonal influenza vaccination has been linked to lower NA responses, or no NA responses, in healthy adults[54]. High-dose vaccination, which often but not consistently includes increased NA protein[53–55], has been shown to induce an anti-NA response in a higher proportion of older adult vaccinees[55], as has been observed in mice[53]. While a significant

N2-specific titer enhancement was not observed in this study, significant differences were observed in the N2-specific humoral immune response across cases and controls both in neutralizing and non-neutralizing functions, arguing for a critical role for NA, not just HA, -specific functional immunity in older adults, in concordance with the recent literature[56–58]. Due to the limitations of the study population and small size, we were unable to distinguish whether the binding to precise NAs and HAs may represent antigen binding (Fab) correlates of protection. Instead, here we observed that a combination of broad HA and NA-specific antibody functions and FcR binding profiles, that included circulating forms of the virus, were selectively enriched in individuals that resisted infection. These data suggest that future vaccines able to drive both functional immunity to HA and NA may synergistically improve immunity in older adults. However, future larger correlates studies, linking Fab-specificities with Fc-functionality may ultimately provide mechanistic insights into the precise epitope-specific functional correlates of immunity against influenza.

The observed diminished ADCC activity of "aging" NK cells builds on previous literature in older adults, demonstrating NK cell anergy, particularly in the setting of cytomegalovirus (CMV) infection[40,41]. Interestingly, while this defect was not linked to age or frailty, the loss of NK cell degranulation and activation was closely associated with CD57 surface expression, a well-established marker of biological aging. CD57 is a marker of exhausted lymphocytes, particularly NK cells and T cells. While not a signaling molecule, CD57 is a terminally sulfated glycan carbohydrate that is highly expressed on lymphocytes with chronic immune activation[59], and expression increases with age[60]. Future studies will be required to determine if CD57 expression is linked to increased risk or severity of infection, and how the development of FCGR3A-signaling defects is related to this biomarker. Furthermore, additional future studies across populations may provide enhanced resolution on strategies to selectively tune Fc-glycosylation to account for progressive NK cell age and comorbidity-associated senescence.

While neutralization and HAI are likely to play a critical role in yearly protection from infection, neutralization and HAI alone represent incomplete metrics of immunity, particularly in older adults[6,61]. Unlike human challenge studies, individuals in this study were likely exposed to influenza at variable times following peak immunogenicity[62], resulting in highly variable immune profiles, particularly among the controls. Yet, despite this heterogeneity in exposure, controls exhibited a more consistent multivariate antibody profile compared to cases, marked by the selective enrichment of FCGR3A and NK cell activating signals in controls. Importantly, NK cells contribute to influenza antiviral control via a diverse set of mechanisms, that extend far beyond simple degranulation of perforin and granzyme[63]. NK cells can mediate a wide array of critical antiviral functions including the induction of antiviral states via cytokine release, the recruitment of additional effector cells via the secretion of chemokines, and the induction of target cell lysis via the TNF-related apoptosis-inducing ligand (TRAIL) or Fas ligand (FasL)[64]. Thus, the selective enrichment of HA and NA-specific cytokine activating antibodies in the controls in this study, and not degranulation-biased immune responses, may point to non-granule mediated killing as a key mechanism of protection against influenza. Importantly, depending on the co-ligation of Fc-receptors and additional NK cell receptors, as well as the strength of Fc-receptor binding/signaling by combinations of IgG subclasses/isotypes/Fc-glycans, distinct NK cell functions may be elicited and controlled by future therapeutic and vaccine strategies. Fc-receptor polymorphisms present in the human population also influence the ability of antibodies to drive NK cell activation[65]. Thus, future in vivo dissection of antibody-mediated cellular functions, particularly within the lung, may reveal the precise mechanism(s) by which NK cells contribute to the control and eradication of Influenza.

Given that vaccination remains the most effective approach to prevent influenza infections in older adults[66–68], strategies are urgently needed to improve both HAI and additional functional correlates of immunity in this population[66,67]. Novel strategies including adjuvants and new immunogens able to induce ADCC – at the required level marked by CD57 expression - may complement emerging efforts that show elevated HAI activity and breadth[69]. Future studies will determine whether these novel vaccine technologies, including the newest innovations currently used for SARS-CoV-2 vaccination, will yield superior responses against influenza in high-risk populations by eliciting broadly functional and NK cell-activating polyclonal humoral immune responses.

## Methods

### Samples

100 samples drawn 28 days post vaccination from elderly individuals were selected from a previously published clinical trial (NCT: 01427309)[8]. All participants in the trial provided informed consent. Exclusion criteria, as well as comorbidity information, for this trial can be found in the previous publication[8]. This trial compared the standard 15 μg per HA dose or high-dose 60 μg per HA Fluzone (Sanofi Pasteur) seasonal influenza vaccines[8], so individuals were included who received both doses ($n = 50$ per vaccine type). These individuals were followed for the duration of the flu season and influenza infection status was confirmed by both PCR and culture. This was not a controlled exposure study, so the influenza exposure status of uninfected individuals is unknown. 14 individuals selected for this study were infected with influenza (any subtype), of which 3 received the high-dose vaccine. Controls were matched for age, gender, and comorbidities. This study was not pre-specified in the clinical trial protocol. This manuscript represents a post-hoc analysis. For more detail regarding the study population, see Table S1.

For comparisons of NK cells from younger and older adults, cryopreserved PMBCs were provided by the University of Connecticut Health Center's Center on Aging. The original study was approved by

the UCHC Institutional Review Board. All participants in the study provided informed consent. For more detail regarding the study population, see Table S2.

All studies utilizing human samples reported in this manuscript were additionally approved by the Massachusetts General Hospital Institutional Review Board.

### Systems serology

For a summary of antigens assayed in each systems serology assay, see Table S3.

### Antibody isotyping, subclassing, and FCGR binding

Antigen-specific antibody isotype, subclass, and FCGR binding titers in participant serum were measured using a custom Luminex-based array as previously published[70–72]. Antigens of interest from past, concurrent, and later influenza seasons were coupled to Luminex beads (Luminex Corp.) through carboxyl chemistry. Antigens assayed were: wild type H3 A/Victoria/361/2011, egg adapted H3 A/Victoria/361/2011, H1 stem from H1 A/New Caledonia/20/1999, H1 A/California/07/2009, N1 A/California/07/2009, H1 A/Brisbane/59/2007, H1 A/Chile/1983, H1 A/New Caledonia/20/1999, H3 A/Texas/50/2102, N2 A/Texas/50/2012, H3 A/Brisbane/10/2007, H3 A/Hong Kong/4108/2014, N2 A/Hong Kong/4108/2014, H3 A/Panama/2007/1999, H3 A/Singapore/19/2016, H3 A/Switzerland/9715293/13, HA B/Brisbane/60/2008, HA B/Phuket/3073/2013, HA B/Colorado/06/2017, and ZEBOV GPdTM. All antigens were provided by Sanofi Pasteur except the A/Victoria/361/2011 HAs and H1 stem[73] (a group 2 stem was not available at the time of the analysis), which were provided by the Ragon Protein Production Core Facility, and the ZEBOV GPdTM, purchased from Mayflower Biosciences. Antigens were produced in mammalian cells and provided only with negligible endotoxin contamination for in vitro experiments. Antigen-coated beads were incubated with diluted serum overnight at 4 C, shaking. For IgG1 and FCGR detection, samples were diluted to a final concentration of 1:500. For IgG3, IgA1, and IgM detection, samples were diluted to a final concentration of 1:100. Following immune complex formation, beads were washed and incubated with a PE-labeled detection reagent. For antibody isotypes IgG1, IgG3, IgA1, and IgM, PE-labeled detection antibodies were purchased from Southern Biotech (Catalog Nos./Clone Nos. 9052-09/4E3, 9210-09/HP6050, 9130-09/B3506B4, and 9020-09/SA-DA4 respectively) and used at a dilution of 1:160. For FCGRs, FCGR2A (R), FCGR2B, and FCGR3A (V), proteins were acquired from the Duke University Protein Production Facility and incubated at a 4:1 molar ratio with streptavidin-PE (Prozyme) prior to addition to immune complexes at a final concentration of 1 μg/ml. FCGR allotypes that were most common in our population (95% white) were used. After 1 hour incubation at room temperature, excess detector was washed away and labeled immune complexes resuspended in QSol buffer (Intellicyt). Plates were read using an iQue Screener PLUS with Forecyt v7 software (Intellicyt), and antibody levels quantified by PE median fluorescence intensity. All samples were assayed in duplicate and quality control was performed to ensure correlation between sample runs and adequate signal-to-noise ratios.

### Antibody-dependent cellular phagocytosis (ADCP)

Antigens (wild type H3 A/Victoria/361/2011, egg adapted H3 A/Victoria/361/2011, and N2 A/Texas/50/2012) were biotinylated using ezLink NHS-LC-LC-biotin (ThermoFisher) for 30 minutes at room temperature, cleaned up with Zeba spin columns (ThermoFisher), and coupled to Neutravidin fluorescent beads (ThermoFisher) for two hours at 37 °C. Beads were washed repeatedly to eliminate unbound antigen and contaminants. Antigen-coated beads were incubated with diluted serum samples (1:200 dilution for HA antigens and 1:100 dilution for NA antigen) for 2 hours at 37 °C, washed, and incubated overnight with 25,000/well THP-1 monocytes (ATCC Catalog No. TIB-202) to allow for phagocytosis. Cells were washed to remove unphagocytosed immune

complexes, then fixed. Plates were read using an iQue Screener PLUS with Forecyt v7 software (Intellicyt). Results are reported as phagocytic scores, which were calculated by multiplying the percentage of monocytes that had undergone phagocytosis by the geometric mean fluorescence intensity of bead-positive monocytes (a proxy for the number of beads phagocytosed by each cell), then dividing by 10,000. All samples were assayed in two independent replicates and quality control was performed to ensure correlation between sample runs and adequate signal-to-noise ratios.

### Antibody-dependent neutrophil phagocytosis (ADNP)

ADNP was assayed as described in ref. 74. Fluorescent beads were coupled to antigens as for ADCP, and incubated with diluted serum samples (1:50 dilution for HA antigens and 1:25 dilution for NA antigen) for 2 hours at 37 °C, washed, and incubated with healthy donor white blood cells isolated by ACK lysis for 1 hour at 37 C. Cells were washed to remove unphagocytosed immune complexes, then stained with CD66b (clone G10F5; BD Catalog No. 561649 or Biolegend Catalog No. 305111) at a dilution of 1:100 to positively identify neutrophils. Cells were fixed and plates were read using an iQue Screener PLUS with Forecyt v7 software (Intellicyt). Results are reported as phagocytic scores, which were calculated by multiplying the percentage of neutrophils that had undergone phagocytosis by the geometric mean fluorescence intensity of bead-positive neutrophils, then dividing by 10,000. All samples were assayed using two independent white blood cell donors and quality control was performed to ensure correlation between sample runs and adequate signal-to-noise ratios.

### Antibody-dependent dendritic cell phagocytosis (ADDCP)

Dendritic cells were derived from blood monocytes isolated from buffy coats (Massachusetts General Hospital Blood Donor Center) by CD14 positive selection (Miltenyi). Following isolation, cells were differentiated in MoDC media (Miltenyi) for 6–7 days. Antigens (wild type H3 A/Victoria/361/2011, egg adapted H3 A/Victoria/361/2011, and N2 A/Texas/50/2012) were coated onto fluorescent beads (ThermoFisher) via carboxy chemistry. Antigen-coated beads were incubated with diluted serum (1:50 dilution for HA antigens and 1:25 dilution for NA antigen) for 2 hours at 37 °C to form immune complexes. Beads were washed and incubated with differentiated DCs for 4 hours at 37 C. DCs were then fixed and stained for cell surface activation makers (HLA-DR (Biolegend Catalog No. 307604/Clone No. L243), CD86 (BD Catalog No. 561128/Clone No. 2331), and CD83 (Biolegend Catalog No. 305330/Clone No. HB15e) all at 0.5 µl/well). Plates were read using an iQue Screener PLUS with Forecyt v7 software (Intellicyt). Results are reported as phagocytic scores, which were calculated by multiplying the percentage of DCs that had undergone phagocytosis by the geometric mean fluorescence intensity of bead-positive neutrophils, then dividing by 10,000, and as percentage of cells that are positive for activation markers. All samples were assayed using two independent DC donors and quality control was performed to ensure correlation between sample runs and adequate signal-to-noise ratios.

### Antibody-dependent complement deposition (ADCD)

ADCD was assayed as described in ref. 75. Fluorescent beads were coupled to antigens as for ADCP, and incubated with diluted serum samples (1:25 dilution for HA antigens and 1:12.5 dilution for NA antigen) for 2 hours at 37 °C, washed, and incubated with purified guinea pig complement (Cedarlane) for 20 minutes at 37 °C. Beads were washed with EDTA-containing buffer, then incubated with anti-C3 fluorescent detection antibody (MP Biomedical Catalog No. 0855385) at 1:100 dilution for 15 minutes at room temperature. Beads were washed and read using an iQue Screener PLUS with Forecyt software (Intellicyt). Results are reported as geometric mean fluorescence intensity of anti-C3 on beads. All samples were assayed in two

independent replicates and quality control was performed to ensure correlation between sample runs and adequate signal-to-noise ratios.

### Antibody-dependent NK cell activation (ADNKA)

NK cells were isolated from healthy donor buffy coats (Massachusetts General Hospital Blood Donor Center) using the RosetteSep NK cell enrichment kit (StemCell). Antigens (wild type H3 A/Victoria/361/2011, egg adapted H3 A/Victoria/361/2011, and N2 A/Texas/50/2012) were adsorbed onto 96-well ELISA plates (ThermoFisher), and plates were blocked in 5% bovine serum albumin (Sigma) in PBS (Corning). Coated plates were then incubated with diluted serum samples (1:25 dilution for HA antigens and 1:12.5 dilution for NA antigen) for 2 hours at 37 C, washed, and incubated with purified donor NK cells for 5 hours at 37 C. The NK cells were then removed from antigen-coated plates, fixed, permeabilized, and stained for both cell surface and intracellular markers (CD3 (0.25 µl/well), CD16 (1 µl/well), CD56(1 µl/well), CD107a (2.5 µl/well), MIP-1β (1 µl/well), IFN-γ (3 µl/well); BD Biosciences Catalog Nos./Clone Nos. 558117/UCHT1, 557758/3G8, 557747/B159, 555802/H4A3, 550078/D21-1351, 340449/25723.11, respectively). Cells were analyzed with an iQue Screener PLUS with Forecyt software (Intellicyt). Results are reported as the percent of NK cells (CD3−, CD56/CD16+) positive for each activation marker (CD107a, IFN-γ, and MIP-1β). All samples were assayed using two independent NK cell donors and quality control was performed to ensure correlation between sample runs and adequate signal-to-noise ratios.

Because PBMCs were not available from the vaccine participants, nor were PBMCs collected for younger individuals, PBMCs from a separate previously established cohort focused on defining immunologic age-related differences in influenza-immunity[42,76] were used to compare NK cells from younger and older adults. CD16 antibody (Biolegend Catalog No. 302004) was adsorbed onto 96-well ELISA plates (ThermoFisher), and plates were blocked in 5% bovine serum albumin (Sigma) in PBS (Corning). PBMCs were rapidly thawed, counted, and added to the coated plates, and incubated for 5 hours. The cells were then removed from coated plates, fixed, permeabilized, and stained for live/dead, cell surface, and intracellular markers (CD107a (BD Catalog No. 555802/Clone H4A3 /2.5 µl/well), CD3 (BD Catalog No. 563546/ Clone UCHT1/0.25 µl/well), CD56 (BD Catalog No. 562289/Clone B159/1 µl/well), CD57 (BD Catalog No. 563896/Clone NK-1/1 µl/well), CD14 (BD Catalog No. 563079/Clone MφP9/1 µl/well), CD69 (BD Catalog No. 563835/Clone FN50/1 µl/well), MIP-1b (BD Catalog No. 550078/Clone D21-1351/1 µl/well), CD16 (Biolegend Catalog No. 302006/Clone 3G8/1 µl/well), KIR (Biolegend Catalog Nos. 339512 and 312610/Clones HP-MA4 and CS27/0.5 µl/well each), NKG2D (Biolegend Catalog No. 320832/Clone 1D11/0.5 µl/well), NKp46 (Biolegend Catalog No. 331932/Clone 9E2/0.5 µl/well), Perforin (Biolegend Catalog No. 308130/Clone dG9/1 µl/well), NKG2A (R&D Systems Catalog No. FAB1059S/Clone 131411/0.5 µl/well), and NKG2C(R&D Systems Catalog No. FAB138A/Clone 134591/4 µl/well)). Cells were analyzed on a BD Fortessa 5-laser cytometer with Diva software. Flow cytometry files were analyzed with Flowjo.

### Antibody glycan analysis

Serum samples were heat inactivated at 56° C for one hour, then spun down at >20,000 g to remove debris. Samples were pre-cleared by incubation with streptavidin-coated magnetic beads (NEB) for 1 hour, rotating. Antigens (wild type H3 A/Victoria/361/2011 and N2 A/Texas/50/2012) were biotinylated for ADCP and coupled to streptavidin-coated magnetic beads. Antigen-coated beads and pre-cleared samples were incubated for one hour at 37 C, rotating. Samples were then washed three times and resuspended in IDEZ (NEB) to cleave at the Fc-Fab hinge. Fc glycans were isolated and labeled using the GlycanAssure kit (ThermoFisher) and APTS-labeled glycans were analyzed by capillary electrophoresis on a 3500xL genetic analyzer with GlycanAssure software (ThermoFisher Applied Biosystems, v2). Pre-labeled libraries

are used as references. Results are reported for each identified peak as percentage of total isolated glycans, and samples without identifiable traces are excluded from the analysis. Peaks are combined into gly-coform groups for visualization and statistical testing (e.g. combining all sialylated glycoforms).

## Statistical and machine learning analysis

Univariate analyses and visualizations were completed in Prism (Graphpad, v9). Statistical tests are detailed in figure legends.

All functional data was averaged across primary cell donors, and no phenotypic and functional differences were observed across neutrophil, NK, or DC donors.

HA comparison heat map and correlation heatmaps were created in R (v3.5.1) using the corrplot package (https://cran.r-project.org/web/packages/corrplot/). Chord diagram was also created in RStudio using the circlize package (https://cran.r-project.org/web/packages/circlize/index.html). Function heatmaps were created in JMP Pro 14 (SAS) using the hierarchical clustering function.

Multivariate and machine learning analyses and visualizations were completed in Matlab R2020a (Mathworks). For Elastic Net-Partial Least Squares analysis of influenza-infected compared to uninfected individuals, a dataset containing antigen-specific antibody isotypes, FCGR binding, neutralization, and antibody-dependent functional data for all 100 study subjects was used. Data were curated prior to multi-variate analysis by removing all below or at-background datapoints, as determined by both PBS controls and EBOV-specific (negative control antigen) background levels. Missing datapoints were imputed using K-nearest neighbor and data were normalized using z-scoring to remove spurious comparisons based on different readouts. Elastic Net machine learning was performed in a fivefold cross-validation framework. The Elastic Net lambda coefficient was chosen within this cross-validation framework. A minimum of five antibody features were selected in each replicate. Random size-matched and label-permuted datasets were compared for each model within the cross-validation framework. This analysis was based on a previously published method[30]. Code for this analysis is provided in Supplementary Software 1.

Networks were plotted in Cytoscape v3.8.1 using pairwise correlation data calculated in Matlab.

## Reporting summary

Further information on research design is available in the Nature Portfolio Reporting Summary linked to this article.

# Data availability

The data generated in this study are provided in the Supplementary Information/Source Data file. Source data are provided with this paper.

# Code availability

The code files generated in this study are provided in Supplementary Software 1.

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

## Acknowledgements

The authors would like to dedicate this paper to Dr. Janet McElhaney and Dr. Todd Suscovich, both who sadly passed prior to the publication of this study, but who had generously contributed their immense knowledge and valuable time, and from whom this paper greatly benefited. The authors would like to thank C. Luedemann, M. Davis, and S. Taylor for their administrative assistance. This work was supported by the following: Sanofi Pasteur, Inc, the Ragon Institute, the Samana Cay Massachusetts General Hospital scholar program (G.AG.A..), and NIH grants R37AI080289 (G.A.), R01AI146785 (G.A.), R01AI153098 (G.A.), U19AI42790-01 (G.A.), U19AI135995-02 (G.A.), U19AI42790-01 (G.A.), P01AI1650721 (G.A.), U01CA260476 – 01 (G.A.), CIVIC75N93019C00052 (G.A.), as well as NIH T32 AI007245 (CMB). We would also like to acknowledge Harvard CFAR for ongoing support through P30 AI060354-02 (G.A.).

## Author contributions

C.M.B., D.L., H.K., V.L., S.S., and G.A. designed the research study. I.D.B. and V.L. participated in the original clinical study from which samples were drawn. S.J., C.V., and G.K. provided older adult NK cells and designed that experiment together with C.M.B. and G.A. C.M.B., J.S.B., M.S., and A.S.Y. conducted experiments and acquired data. C.M.B. analyzed data. C.M.B. and G.A. wrote the manuscript. All authors contributed to the final version of the manuscript.

## Competing interests

G.A. is a founder and equity holder of Seromyx Systems, a company developing a platform technology that describes the antibody immune response. C.M.B. is an employee and equity holder of Leyden Labs, a company developing pandemic-prevention therapeutics. G.A. is an employee and equity holder in Moderna, a company developing mRNA therapeutics and vaccines. H.K. is an employee and equity holder in SK Biosciences, a vaccine developer. I.D.B., V.L., and S.S. are employees of Sanofi Pasteur, Inc. The remaining authors declare no competing interests.
