## [Peer Review File · Nature Communications]

Antibody-mediated NK cell activation as a correlate of immunity against influenza infectionReviewers' Comments:

Reviewer #1:

Remarks to the Author:

The manuscript submitted by Boudreau et al. gives a comprehensive insight into NK cell-shaping humoral immune responses upon vaccination of older adults against influenza. The authors compared immune characteristics in individuals vaccinated with standard vs. high dose vaccines that later became infected with influenza and those harboring efficient protective immunity. Here, they could identify NK cell activating antibodies against influenza H3N2 antigens as predictors for good vaccine efficacy. In a second study comparing the in vitro-activation of NK cells, derived from younger and aged individuals, by an ADCC-inducing antibody, the authors could identify the senescence marker CD57 as a negative correlate of NK cell activatability. Based on the presented data, Boudreau et al. propose that next generation vaccines designed for the elderly should selectively boost NK cell activating antibodies in order to improve vaccine efficacy. In this line, they further highlight the importance of protective ADCC signals against HA as well as NA proteins.

The manuscript is well written and structured adequately with a clear story line. The systems serology and mathematical modelling approaches allow an unbiased analysis of multivariate innate immune functions thereby facilitating a deconvolution of a multidimensional dataset. This led to the identification of a target for improved vaccination approaches.

A few points should be addressed prior to publication of the manuscript:

- Line 95: How were the asymptomatic individuals identified? Was routine testing performed in all participants?
- Regarding the older adults enrolled in the FIM12 efficacy trial: How were the 100 participants analyzed here chosen from the trial? Which criteria were used? Is there anything known about their medical history (e.g. chronic infections?) and previous influenza vaccinations (e.g. frequency, kind of vaccine)? This information would be valuable and could help to further understand the immunological differences observed in the elderly. If this data is available in another manuscript, this should be clearly indicated. If such data is not available, this should be mentioned. Especially in the elderly, the immune system is shaped by previous encounters with pathogens (e.g. theory of original antigenic sin) and a short paragraph highlighting this point should be included in the discussion.
- The fact that for the elderly trial no baseline (day 0) samples were available should be described in the beginning of the results part in order to clarify that all profiles include vaccine-induced as well as "natural" occurring immune responses (due to previous infections or vaccinations) as stated in Line 118-119. This explanation should also include the fact that especially elderly have encountered influenza through vaccination or natural infection several times before the vaccination as part of the clinical trial (see comment above). Thus, in elderly, an influenza vaccination can be regarded as a booster rather than a "new" vaccination and the induced response will be shaped by the previous encounters. This should be additionally addressed in the discussion. It is especially important to mention that no baseline data is available when it comes to the description of "vaccine-induced" effects.
- The time point of sampling (day 28 post vaccination) should be included in the legend for figure 1.
- Line 97-102: "In line with previous studies pointing to an incomplete association between HAI and protection, individuals who did not become infected during the study period (controls) trended toward higher post-vaccination and pre-infection HAI titers towards the circulating H3N2 strain compared to the individuals who later became infected (cases), although this difference was not significant ($p = 0.075$, Figure 1A)." If controls had higher HAI titers pre-infection as compared to cases, how is this pointing to an incomplete association between HAI Titers and protection? Please comment.
- Were HAI Titers against all vaccine antigens tested? It would be important to show the response against all vaccine antigens. If there were only differences observable for H3N2, do you have an explanation for this specificity?

- While the HAI Assay gives information on the quantity of antibodies, it does not yield any information on their actual functionality. Was functional testing (e.g. MN assay) performed in the context of the study? This might have given additional valuable information. Please comment.
 - Please specify what data set (e.g. derived from Luminex analysis) was used for the in depths analyses, e.g. the LASSO-analysis or the analysis of stalk- or NA-specific antibody profiles.
 - Figure S3: There are lines in graphs B and K. Were there supposed to be asterisks indicating significant differences? The Y-axes are all shown in scientific numbers, but not the graph in F).
 - The description of figure S3 I-L is a little confusing and should be refined with regard to the order of the graphs vs. its description as well as the description vs. the labels of the graphs.
 - Line 199-201: "Interestingly, bisecting-n-acetyl glucosamine (GlcNAc) and sialylation (S) were also associated with enhanced NK cell cytokine secretion, [...]" The respective figure 3A shows a negative correlation between GlcNAc and Di-Sialic Acid with ADNKA IFN- γ . Please comment.
 - In figure 4E it seems like there is at least a trend towards enhanced frequencies of CD107a+ NK cells upon vaccination with the high dose vaccine. This might be a hint that high dose vaccines induce a stronger NK cell activation as compared to the regular dose vaccines. Please comment.
 - Line 250-252: "While NK cell activation was directly linked to age (for CD107a Spearman's $R = -0.2347$, ns; for MIP-1b Spearman's $R = 0.0153$, ns), [...]" "Conversely, no association with NK cell function or phenotype was observed with BMI or clinical frailty." Where is this data shown?
-
- Line 26: A word is missing. (...adults that were monitored for influenza infection...)
 - Line 126-129: Here it would be helpful for the reader to include the respective markers shown in figure one in order to follow the reasoning.
 - Line 160: There is one word too much ("did").

Reviewer #2:

Remarks to the Author:

Boudreau et al present a serological functional correlates of protection study in older adults from standard vaccination, impacts of high dose vaccination, and then ex vivo representative NK cell function with age impacts. Some clarification is need in the text mainly for readership, and technical details rather than further experiments. In the first part of the study, peak immunogenicity is compared between infected cases and control post standard vaccination when a H3N2 mismatch occurred (however infection subtype is not given?), does this reflect an individuals capacity to respond to vaccination or their prior immunity? These signatures for sero-protection were then compared versus high dose inactivated vaccines, and not augmented by increased dose. The ability to recruit NK cells from older versus younger direct ex vivo donors was compared, independent of the vaccine studies, where younger donor cells were used for different immune serum. Pairing immune serum with functional assays when there is multiple timepoints would be no reasonable feat, but was it tested in the ex vivo donor study as paried serum or inverted for age- i.e. old cells with young serum and vice versa?

Experimental

1. There is a disconnect between the direct ex vivo NK cell function to non-specific CD16 activation, and the earlier influenza infection and vaccination study. Do the findings in figure 5 (CD57 exhausted NK cells are less functional) affect the data from other figures which use healthy younger buffy coat NK cells with serum from older adults? With reduced functional NK cells in older adults, is the measured NK cell activity truly occurring in vivo and contributing to protection? Do authors have PBMCs from participants to demonstrate NK cell CD57 expression as a further predictor of infection?
2. Line 3964 – why use the low affinity Fc γ R2a receptor?
3. Phagocytosis assays - what is the background level of non-antibody dependent phagocytosis and what is used as a negative control? Are these assays representative of phagocytosis in vivo as beads

are presumably easier to phagocytose than infected cell immune complexes?

4. ADNP - Is CD66b sufficient to identify neutrophils from whole blood as is also on eosinophils? Why focus on phagocytosis and not other antibody dependent neutrophil functions? Would neutrophil degranulation have similar importance to NK cell?

5. All cell assays – are significant differences in surface markers or responses seen between independent healthy blood donors cells used?

Minor comments:

1. Abstract, context, “subsequent season”, when was this study conducted and which subtypes, which vaccines was used? High dose is later mentioned. What is the age range of the older adults? (65-90yo)? Further brief details in the abstract would help for easy context.

i.e. standard seasonal inactivated influenza virus vaccine in 2012/2013 NH in the UK?

2. Abstract: “peak immunogenicity” add “of their vaccine response”, also intro line 75. What timepoint does this correspond to? i.e. 7 days post vaccination or 28 days post vaccination? Given there is no pre vaccination timepoint, the peak isn’t really known. As ADCC antibodies likely peak before neutralisation/HAI responses (Based on Jegaskanda macaque studies by infection 10.1128/JVI.03030-12, but not by vaccination 10.1128/JVI.01666-13). What is the time interval between vaccination and infection in cases? And when is ‘peak immunogenicity’ measured? (finally given in methods at line 344, 28 days post vaccination)

3. Abstract: what is meant by “orthologous cohort”, an independent cohort w the same vaccine?

4. Abstract: CD57 is not an ageing marker “the age-dependent NK cell marker CD57”, but an activation maker w cytotoxic potential for NK cells. Reword here. In results line 247, referred to as “a marker of senescence”.

5. Abstract “The current vaccination strategy for older adults, high dose vaccination”, this is not the current vaccination in some countries, where standard or adjuvanted vaccines are used instead in older adults. Reword.

6. Introduction, line 44 “the United States”, reword as international readership.

7. Intro, line 53, which antigen was mismatched? H3N2? H3N2 Tx12 versus HK14?

8. Intro line 57, “extra-neutralizing” reword, suggest : antibody correlates beyond neutralising function.

9. Results, line 91/92, add 2013.

10. Results, line 93, “strain A H3N2” further details needed, NCBI/GenBank ID? How many egg adapted mutations were in the vaccine HA versus the circulating strain? Did both the standard and high dose vaccines contain egg derived antigens of the same strain? Some details would help for context.

11. Figure 1c, it is unclear what is measured in ‘breadth’? is this different H3N2 viruses? What is measured here? Table S3 should be referred to.

12. Figure S2d, the case data should be shown here also, it is meaningless as control data alone.

13. What is the H1-stem protein that is being used? A chimeric cH6/1? Headless HA-mini stem? As the cases become H3N2 infected, would a group 2 stalk be more relevant to measure as group 1 vs group 2 HA-stalk responses can be exclusive of each other.

14. Figure 5, can the frequency of NK cells, expression of CD57 and CD16 be shown as a function of age, as a continuous variable? Are correlations in figure 5 def, based on both younger and older NK cells, does the strength of the relationship change w age? NK cell function is also time dependent from collection (<12 hours), what time post bleed were these cells tested and were they freeze thawed?

15. Figure 2 – Authors say that due to heterogeneity, multivariate analysis would more likely predict the differences between cases and controls, as shown here – but did authors do univariate analyses for these ADCC related binding and functions? And if so were any NK cell activities independent predictors of protection? Is this necessary for the conclusion sentence on line 1874?

16. Table S1, this cohort information is missing age, gender, prior vaccine history data.

17. Line 1264 – Please include H3N2 and H1N1 cases in Table S1. Were all cases included in analysis, or just H3N2?

Reviewer #3:

Remarks to the Author:

The manuscript entitled "Antibody-mediated NK cell activation as a correlate of immunity against influenza infection" submitted by Boudreau et al. employs systems serology to dissect markers of protective immune responses upon influenza vaccination. Comparing serum samples from vaccinees who later on got infected with influenza or did not get infected, the authors conclude that activation of NK cells plays a major role in immune protection. The study provides very valuable insights regarding Fc-mediated effector functions in protection from viral infection. I find the conclusion of the data pointing towards non-granule mediated killing as a mechanism especially interesting also related to other viral infections. Overall, the study is of high quality and the conclusions are well supported by the presented experimental data.

I, however, would kindly ask the authors to elaborate on the following points:

1. Figure 1: I am a bit confused by the different abbreviations. While the main text mentions "HA-specific Fc profiles" the figure legend refers to H3 and HAI antibodies. For a non-specialist it is unclear if that is the same. Furthermore, why were detailed results shown for H3-antibodies but not HAI antibodies? If there were no notable correlations this could be mentioned specifically. Of note, the results section for figure 2 additionally talks about NA-antibodies which are also not mentioned in figure 1.

2. Line 151-169: How do the authors imagine the do novo response to non-H3 vaccine strains to be initiated in absence of infection with the respective virus strains (subjects primarily infected with H3N2)? If these subjects have been infected with H1 virus before, I personally would not call the response "de novo". Could it be some form of bystander activation?

3. Concerning interpretation of Fc glycosylation results: The authors mention increased fucosylation and increased di-sialylation in cases over controls. In fig. 3b one can see that a prominent part of the control group show comparable fucosylation as the case group. This suggests that fucosylation might not solely account for the observed differences and I thus find the statement in lines 205-207 a bit strong. Was there a correlation of NK cell activation (and other measured parameters) with the level of fucosylation in individual samples? What about correlation of distinct glycosylation profiles within donors e.g. fucosylation and glycosylation? If possible, providing primary data of these analyses would be helpful in terms of getting further insights into the role of distinct glycosylation patterns for human IgG activity.

4. With respect to differences observed in NK cell activation in vitro and more importantly protection of vaccinees I am also wondering if the authors had the opportunity to genotype for the FcgRIIIa-158V/F polymorphism as this has previously been suggested to impact IgG binding. Even if not, this could be a point to add to the discussion.

5. In line 329, the authors propose the induction of specific NK cell functions by targeting specific receptors. This is indeed a promising perspective for future vaccines and/or therapeutics but I find the word "can" again a bit strong given our lack of knowledge on the interplay of all those receptors. I would kindly suggest to rephrase.

Last but not least I am curious about a couple of methodological details:

- Why were serum samples diluted differently for ADCP and ADNP assays? Along those lines, did the authors account for heterogeneity in antigen-specific IgG levels for serum dilutions or when analysing the data? Alternatively, were beads checked for saturation with donor IgG? If not, it might be more difficult to dissect the inflammatory potential of IgG in distinct samples independent of its concentration.

- Why was the ADCP assay done separately with THP-1 cell line instead of taking that data from the ADNP assay that not only included neutrophils but also monocytes upon ACK lysis of blood samples?

Line 26: "were monitored influenza infection" seems like a word is missing

Line 50: correlate instead of correlates?

Line 78: split sentence after "influenza"

Line 166: spelling error in specific

Reviewer #4:

None

Antibody-mediated NK cell activation as a correlate of immunity against influenza infection

RESPONSES TO REVIEWER COMMENTS

The authors would like to open by thanking all three reviewers for their time and thoughtful comments. We heartily appreciate their efforts to improve our manuscript. We believe that the manuscript has been strengthened by the inclusion of the reviewer's suggested changes.

Reviewer #1 (Remarks to the Author):

The manuscript submitted by Boudreau et al. gives a comprehensive insight into NK cell-shaping humoral immune responses upon vaccination of older adults against influenza. The authors compared immune characteristics in individuals vaccinated with standard vs. high dose vaccines that later became infected with influenza and those harboring efficient protective immunity. Here, they could identify NK cell activating antibodies against influenza H3N2 antigens as predictors for good vaccine efficacy. In a second study comparing the in vitro-activation of NK cells, derived from younger and aged individuals, by an ADCC-inducing antibody, the authors could identify the senescence marker CD57 as a negative correlate of NK cell activatability. Based on the presented data, Boudreau et al. propose that next generation vaccines designed for the elderly should selectively boost NK cell activating antibodies in order to improve vaccine efficacy. In this line, they further highlight the importance of protective ADCC signals against HA as well as NA proteins.

The manuscript is well written and structured adequately with a clear story line. The systems serology and mathematical modelling approaches allow an unbiased analysis of multivariate innate immune functions thereby facilitating a deconvolution of a multidimensional dataset. This led to the identification of a target for improved vaccination approaches.

A few points should be addressed prior to publication of the manuscript:

- Line 95: How were the asymptomatic individuals identified? Was routine testing performed in all participants?

We are grateful to the reviewer for identifying this use of imprecise language. Routine testing was not performed on all participants, however any respiratory illness (not only influenza-like illness) triggered the testing of participants. These individuals were not asymptomatic in the sense that they were healthy; rather, they displayed respiratory symptoms that did not rise to the level of influenza-like illness.

Three definitions of respiratory illness could trigger testing:

1. Respiratory illness: the occurrence of one or more of the following: sneezing, nasal congestion or rhinorrhea, sore throat, cough, sputum production, wheezing, or difficulty breathing

2. Protocol-defined influenza-like illness: sore throat, cough, sputum production, wheezing, or difficulty breathing, concurrent with one or more of the following: temperature above 37.2°C, chills, tiredness, headaches, or myalgia
3. CDC-defined influenza-like illness: cough or sore throat, concurrent with a temperature above 37.2°C

Line 95 has been changed to reflect this distinction: that these individuals suffered respiratory or systemic symptoms.

- Regarding the older adults enrolled in the FIM12 efficacy trial: How were the 100 participants analyzed here chosen from the trial? Which criteria were used? Is there anything known about their medical history (e.g. chronic infections?) and previous influenza vaccinations (e.g. frequency, kind of vaccine)? This information would be valuable and could help to further understand the immunological differences observed in the elderly. If this data is available in another manuscript, this should be clearly indicated. If such data is not available, this should be mentioned. Especially in the elderly, the immune system is shaped by previous encounters with pathogens (e.g. theory of original antigenic sin) and a short paragraph highlighting this point should be included in the discussion.

We thank the reviewer for the opportunity to clarify what is known about our study population. Information on previous influenza vaccination (prior to six months before study enrollment) was not collected. For inclusion in the parent trial, the exclusion criteria were "Participants were excluded if they had: history of Guillain-Barré syndrome, systemic hypersensitivity or life-threatening reaction to the study vaccines or their components; received influenza vaccination within 6 months prior to enrollment; thrombocytopenia, bleeding disorder, or had received anticoagulants contraindicating intramuscular vaccination; dementia, any cognitive condition, alcohol abuse, or drug addiction at a stage that could interfere with study compliance; or were: participating (or had participated) in another interventional study within 4 weeks preceding enrollment; investigators or their employees or immediate family members; or deprived of freedom."¹ All samples with adequate remaining volume for analysis from individuals that tested positive for influenza infection during the study period were included in this sample set. Non-infected samples were selected by matching for age, sex, and vaccine type; an equal number of regular dose and high dose samples were selected. CMV serostatus, a known contributor to immune function in older ages, was not collected for this population. These clarifications been added to the methods, and a mention of these implications included in the discussion.

- The fact that for the elderly trial no baseline (day 0) samples were available should be described in the beginning of the results part in order to clarify that all profiles include vaccine-induced as well as "natural" occurring immune responses (due to previous infections or vaccinations) as stated in Line 118-119. This explanation should also include the fact that especially elderly have encountered influenza through vaccination or natural infection several times before the vaccination as part of the clinical trial (see comment above). Thus, in elderly, an influenza vaccination can be regarded as a booster rather than a "new" vaccination and the induced response will be shaped by the previous encounters. This should be additionally addressed in the discussion. It is

especially important to mention that no baseline data is available when it comes to the description of “vaccine-induced” effects.

We thank the reviewer for pointing out this key clarification to the interpretation of our results. As suggested by the reviewer, this clarification has been added to the first paragraph of the results and to the discussion.

- The time point of sampling (day 28 post vaccination) should be included in the legend for figure 1.

We appreciate the reviewer’s suggestion; this information has been added to the figure legend.

- Line 97-102: “In line with previous studies pointing to an incomplete association between HAI and protection, individuals who did not become infected during the study period (controls) trended toward higher post-vaccination and pre-infection HAI titers towards the circulating H3N2 strain compared to the individuals who later became infected (cases), although this difference was not significant ($p = 0.075$, Figure 1A).” If controls had higher HAI titers pre-infection as compared to cases, how is this pointing to an incomplete association between HAI Titers and protection? Please comment.

We appreciate the opportunity to clarify our analysis. While the controls tended towards higher HAI titers pre-infection, the difference was not sufficient to reach statistical significance either in our study subset (Figure 1A) or in the larger clinical trial¹. Furthermore, over 90% of total vaccinees had HAI titers of or greater than the amount considered to be protective ($>1:40$)², and the ranges of the two groups were similar (Figure 1A). Thus, while the contribution of HAI to protection cannot be disregarded – and will be emphasized – it is possible that HAI alone is insufficient to provide protection, which provided the motivation for this study. We have now clarified this point in the manuscript.

- Were HAI Titers against all vaccine antigens tested? It would be important to show the response against all vaccine antigens. If there were only differences observable for H3N2, do you have an explanation for this specificity?

We thank the reviewer for bringing up this missing data. HAI titers were measured against H1N1 (A/California/07/2009), H3N2 (A/Victoria/361/2011), and influenza B (B/Texas/6/2011). We chose to focus on H3N2 in this manuscript because it was the leading cause of infections among our cohort. To add additional context, we have added HAI information for the other two strains to Figure S1, and for convenience, have reproduced it here.

Figure S1 (excerpted). Serum samples from day 28 post-vaccination were analyzed for HAI, influenza-specific antibody levels, and influenza-specific antibody-mediated innate immune functionality. Dot plots show HAI titers for controls (n = 86) who remained uninfected and cases (n = 14) who became infected during the season against H1N1 A/California/07/2009 and B/Texas/6/2011. Differences between controls and cases were not significant.

- While the HAI Assay gives information on the quantity of antibodies, it does not yield any information on their actual functionality. Was functional testing (e.g. MN assay) performed in the context of the study? This might have given additional valuable information. Please comment.

We appreciate the reviewer's request for more detailed information; however, additional tests of neutralization were not performed in this study. For future studies, we agree that more direct measures of neutralization (MN) or protectiveness (in vivo studies) would be informative and have added this point to the manuscript limitations.

- Please specify what data set (e.g. derived from Luminex analysis) was used for the in depths analyses, e.g. the LASSO-analysis or the analysis of stalk- or NA-specific antibody profiles.

We thank the reviewer for their question. In-depth multivariate analyses were conducted using the complete antibody feature dataset, comprising Fc-mediated innate immune functions by HA- and NA-specific antibodies, Luminex FcR binding levels against all tested antibody specificities, Luminex antibody isotype levels against all tested antibody specificities, HAI, and NAI data. This has been clarified in the results section, and the complete list of features is available in the data .zip file attached to this submission.

- Figure S3: There are lines in graphs B and K. Were there supposed to be asterisks indicating significant differences? The Y-axes are all shown in scientific numbers, but not the graph in F).

We thank the reviewer for pointing out these typographical errors. It is unclear why the asterisks did not show up for the reviewer as they are visible to the authors; however, the image format has been edited to attempt to rectify this issue.

- The description of figure S3 I-L is a little confusing and should be refined with regard to

the order of the graphs vs. its description as well as the description vs. the labels of the graphs.

We appreciate the reviewer's identification of this confusing legend. The legend and figure have been updated for increased readability.

- Line 199-201: "Interestingly, bisecting-n-acetyl glucosamine (GlcNAc) and sialylation (S) were also associated with enhanced NK cell cytokine secretion, [...]". The respective figure 3A shows a negative correlation between GlcNAc and Di-Sialic Acid with ADNKA IFN- γ . Please comment.

We thank the reviewer for pointing out this error. Lines 199-201 were intended to read "GlcNAc and S were also associated with enhanced NK cell degranulation but reduced NK cell cytokine secretion...". This has been corrected.

- In figure 4E it seems like there is at least a trend towards enhanced frequencies of CD107a+ NK cells upon vaccination with the high dose vaccine. This might be a hint that high dose vaccines induce a stronger NK cell activation as compared to the regular dose vaccines. Please comment.

We thank the reviewer for this insightful comment. We agree with the reviewer's comment, as there is a trend towards a potential dose effect on NK cell activation. However, the differences between regular and high dose vaccinees do not reach significance.

- Line 250-252: "While NK cell activation was directly linked to age (for CD107a Spearman's $R = -0.2347$, ns; for MIP-1b Spearman's $R = 0.0153$, ns), [...]". "Conversely, no association with NK cell function or phenotype was observed with BMI or clinical frailty." Where is this data shown?

We appreciate the opportunity to clarify this data. While these calculations were performed on the included data by the research team, they were not graphically depicted in the manuscript owing to supplemental figure number limitations and a disinclination to show negative data in main text figures. A selection of these graphs have been included here for the reviewers and editors:

Should it be deemed critical to include these plots in the manuscript, the authors would welcome reviewer and editor suggestions of where and how to include them.

- Line 26: A word is missing. (...adults that were monitored for influenza infection...) We thank the reviewer for catching this typographical error. It has been corrected.

- Line 126-129: Here it would be helpful for the reader to include the respective markers shown in figure one in order to follow the reasoning. The relevant markers have been added for opsonophagocytic (ADCP, ADNP), complement fixing (ADCD), and DC activating (ADDCP, HLA-DR, CD83, CD86) antibody functions.

- Line 160: There is one word too much ("did"). We thank the reviewer for catching this typographical error. It has been corrected.

Reviewer #2 (Remarks to the Author):

Boudreau et al present a serological functional correlates of protection study in older adults from standard vaccination, impacts of high dose vaccination, and then ex vivo representative NK cell function with age impacts. Some clarification is need in the text mainly for readership, and technical details rather than further experiments. In the first part of the study, peak immunogenicity is compared between infected cases and control post standard vaccination when a H3N2 mismatch occurred (however infection subtype is not given?), does this reflect an individuals capacity to respond to vaccination or their prior immunity? These signatures for sero-protection were then compared versus high dose inactivated vaccines, and not augmented by increased dose. The ability to recruit NK cells from older versus younger direct ex vivo donors was compared, independent of the vaccine studies, where younger donor cells were used for different immune serum. Pairing immune serum with functional assays when there is multiple timepoints would be no reasonable feat, but was it tested in the ex vivo donor study as paried serum or inverted for age- i.e. old cells with young serum and vice versa?

Experimental

1. There is a disconnect between the direct ex vivo NK cell function to non-specific CD16 activation, and the earlier influenza infection and vaccination study. Do the findings in figure 5 (CD57 exhausted NK cells are less functional) affect the data from other figures which use healthy younger buffy coat NK cells with serum from older adults? With reduced functional NK cells in older adults, is the measured NK cell activity truly occurring in vivo and contributing to protection? Do authors have PBMCs from participants to demonstrate NK cell CD57 expression as a further predictor of infection? We thank the reviewer for raising this important point. Regrettably, PBMCs were not collected from vaccinees in the cohort used for systems serology analysis.

As described in the manuscript and Table S2, samples collected during a separate study on aging^{3,4} at the University of Connecticut Health Center were used to explore the potential influence of age on NK cell functions that were linked to protection in this study, with the aim of forming a cohesive theory of protective mechanisms against influenza in older adults. We have clarified this point in the manuscript.

To address the reviewer's interesting question about the relevance of antibody-mediated readouts using younger cells to evaluate older individual's antibodies, we have 2 considerations:

- 1) There is a need to define biomarkers that predict protection against influenza to guide our mechanistic understanding of immune control of the virus and/or to help guide vaccine design. In the design of Biomarker assays, developing assays that provide the strongest signal-to-noise separation and largest assay range provides the greatest probability of defining robust biomarkers. To this end, young NK cells have consistently provided the greatest reproducibility and signal-resolution. Using this strategy, the data clearly highlight that *in vivo* in older

adults, individuals that do not experience breakthrough influenza elicit antibodies with a higher propensity to drive NK cell activation.

- 2) *In vivo*, NK cells in the aging population appear to be more functionally attenuated. Thus, these data argue that the need for enhanced ADCC inducing antibodies may not relate to a preferential need to leverage NK cells with age, but to generate antibodies able to overcome NK cell anergy. Whether NK cells then drive influenza control and clearance *in vivo* is an important question that is difficult to address in humans. To address this question, we have now performed a mouse experiment, using antibodies with differential capacities to recruit NK cells. These data clearly highlight, and confirm, that NK cell activating antibodies preferentially provides protection against disease and death. This data has been included below and as supplemental data.

Figure S4. NK-activating Fc mutant antibody increases protection in mice

(A) Flower plots depict H3 X-31-specific functional and Fc-binding profiles of mAb CR9114 Fc mutants where larger petals indicate relative strength of that readout compared to other mutants. (B) Survival curves after C57Bl/6 mice were given monoclonal antibodies (CR9114 WT, N297Q or SDIEALGA) or PBS control by passive transfer at a dose of 4mg/kg, then infected with a 3xLD50 dose of H3N2 reassortant X-31 influenza virus (n=5 per group). Significance tested by Mantel-Cox test of survivorship * p < 0.04. (C) Weight loss curves (percent animal body weight, symbols represent means and error bars represent standard error of the mean) for the same murine infection experiment. Mice were euthanized once they reached 20% (or greater) weight loss.

2. Line 3964 – why use the low affinity FcγR2a receptor?

Here we presume the reviewer was referring to the choice of the R131 allotype in the FcγR2A receptor, as both FcγR2A allotypes are typically considered low affinity receptors⁵. While the H131 allotype shows higher affinity for IgG2 isotype antibodies specifically⁶, affinities for the more functional IgG1 and IgG3 isotypes are unchanged. The R131 allotype was chosen because it is the predominant allotype among White individuals, who comprised a majority of the study (~95%)¹. This point has been noted in the methods section.

3. Phagocytosis assays - what is the background level of non-antibody dependent phagocytosis and what is used as a negative control? Are these assays representative of phagocytosis in vivo as beads are presumably easier to phagocytose than infected cell immune complexes?

We thank the reviewer for the opportunity to clarify our experimental conditions. Background phagocytosis is measured by both no serum (PBS) control samples and influenza A virus negative control serum samples (commercially available from IBL, Inc.). The background level is ensured to be at minimum 3-fold below positive control samples. The beads utilized in these samples have a diameter of 1 μm, which while smaller than most infected cells is larger than an influenza virion, and was chosen to represent a middle phenotype to measure the possibility of both of these conditions in vivo through a single sample-sparing measurement.

4. ADNP - Is CD66b sufficient to identify neutrophils from whole blood as is also on eosinophils? Why focus on phagocytosis and not other antibody dependent neutrophil functions? Would neutrophil degranulation have similar importance to NK cell?

We appreciate the reviewer's questions related to the neutrophil assay. The reviewer is correct that eosinophils may be contaminating our results. However, as eosinophils make up only 1-4% of total white blood cells against the 40-60% neutrophil composition, eosinophils are a minor player in this assay. This consideration has been added to the methods section.

Related to further dissection of neutrophil function, previously we have observed a strong relationship between phagocytosis and degranulation⁷. In other words, degranulation was tightly correlated with phagocytosis, but requires a longer incubation to enable the cells to release granules. Due to the limitation of plasma that we received, we were unable to run this additional readout, and felt that capturing orthogonal readouts, including NK cells, monocytes, DCs, etc. was potentially important for this first exploratory analysis- but agree that further dissection will be of great interest. We have added this point to the discussion.

5. All cell assays – are significant differences in surface markers or responses seen between independent healthy blood donors cells used?

Based on the staining panels used in each assay, there were no baseline/control well differences in cell phenotype and function across donors for each assay. This is a critical initial quality control step in all Systems Serological assays and has been included in the methods section. It is important to note that our panels do not preclude the possibility of additional changes in cells across donors, which is why data is averaged across donors to account for natural cellular variation.

Minor comments:

1. Abstract, context, “subsequent season”, when was this study conducted and which subtypes, which vaccines was used? High dose is later mentioned. What is the age range of the older adults? (65-90yo)? Further brief details in the abstract would help for easy context.

i.e. standard seasonal inactivated influenza virus vaccine in 2012/2013 NH in the UK?

We are grateful for the reviewer’s suggestion to add additional clarity to our abstract.

The abstract has been edited to include cohort details.

2. Abstract: “peak immunogenicity” add “of their vaccine response”, also intro line 75. What timepoint does this correspond to? i.e. 7 days post vaccination or 28 days post vaccination? Given there is no pre vaccination timepoint, the peak isn’t really known. As ADCC antibodies likely peak before neutralisation/HAI responses (Based on Jegaskanda macaque studies by infection 10.1128/JVI.03030-12, but not by vaccination 10.1128/JVI.01666-13). What is the time interval between vaccination and infection in cases? And when is ‘peak immunogenicity’ measured? (finally given in methods at line 344, 28 days post vaccination)

We appreciate the reviewer’s flagging of this issue. We have opted to remove the term “peak immunogenicity” as it complicated, and did not improve, the interpretation of our study.

3. Abstract: what is meant by “orthologous cohort”, an independent cohort w the same vaccine?

We thank the reviewer for the chance to clarify. By orthologous cohort, we meant an independent cohort reflecting the same population (community-dwelling US adults >65 years old). In this case, we have edited the language to be more precise.

4. Abstract: CD57 is not an ageing marker “the age-dependent NK cell marker CD57”, but an activation maker w cytotoxic potential for NK cells. Reword here. In results line 247, referred to as “a marker of senescence”.

We appreciate the opportunity to further dissect CD57 biology. Currently, while many publications still utilize CD57 as a marker of immunosenescence on NK and T cells⁸, there is debate in the field regarding the accuracy of that marker. It is clear from the literature that CD57+ NK cells are mature cells that retain cytolytic activity while displaying reduced proliferation (reviewed in ⁹). To this end, we have clarified both marked sentences to avoid overstating a currently disputed claim in the literature.

5. Abstract “The current vaccination strategy for older adults, high dose vaccination”, this is not the current vaccination in some countries, where standard or adjuvanted vaccines are used instead in older adults. Reword.

We thank the reviewer for this important clarification. This has been corrected to specify that high dose vaccination is currently used for older adults in the United States, where this trial took place.

6. Introduction, line 44 “the United States”, reword as international readership.

We thank the reviewer for their observation that this study is applicable to more than just the United States. Here, however, we cite a specific study and the aging population of the United States as a specific cause of increasing deaths. To reword globally would be overstating the cited study. This idea was part of the motivation for this study, as it took place in the United States.

7. Intro, line 53, which antigen was mismatched? H3N2? H3N2 Tx12 versus HK14? We appreciate the chance to clarify this sentence. During the 2012/2013 season, the H3N2 strain underwent a mutation during vaccine production, which improved the virus's fitness in eggs but resulted in a change in the HA at a critical site for antibody binding. This resulted in lower vaccine efficacy. The text has been clarified to indicate that the H3N2 strain had the mismatch.

8. Intro line 57, "extra-neutralizing" reword, suggest : antibody correlates beyond neutralising function.

We appreciate the reviewer's suggestion. This edit has been made.

9. Results, line 91/92, add 2013.

We appreciate the reviewer's suggestion. This edit has been made.

10. Results, line 93, "strain A H3N2" further details needed, NCBI/GenBank ID? How many egg adapted mutations were in the vaccine HA versus the circulating strain? Did both the standard and high dose vaccines contain egg derived antigens of the same strain? Some details would help for context.

We appreciate the reviewer's request; however, the exact strain was not defined in the testing of these samples. They were tested to differentiate between H1N1, H3N2 and influenza B.

To address the reviewer's additional questions, the egg adapted strain had three amino acid mutations in the HA. Both the standard and high dose vaccines contained egg-derived antigens of the same strain, with the same antigenic mismatch to the circulating strain. This context has been added to the text.

11. Figure 1c, it is unclear what is measured in 'breadth'? is this different H3N2 viruses? What is measured here? Table S3 should be referred to.

We thank the reviewer for noting this critical omission. The columns of the heat map under the heading breadth measure the relative level of an antibody isotype or FcγR binding across all tested influenza antigens (Table S3, Ag-specific Ab isotype titers and FcR binding). The heat map columns are in the order: total IgG, IgG1, IgG3, IgA1, IgM, FcγR2A, FcγR2B, and FcγR3A. Clarification has been added to the figure legend.

12. Figure S2d, the case data should be shown here also, it is meaningless as control data alone.

We appreciate the opportunity to further explain Figure S2D and to correct a typographical error. This figure represents all study participants, not just controls as was erroneously stated. With this figure, we intended to make the point that broad IgG1 levels across the vaccine antigen, circulating antigen, and a heterosubtypic antigen

were present across infection outcomes. We have colored the figure to make the cases vs controls distinction visible so that this figure may be more informative.

13. What is the H1-stem protein that is being used? A chimeric cH6/1? Headless HA-mini stem? As the cases become H3N2 infected, would a group 2 stalk be more relevant to measure as group 1 vs group 2 HA-stalk responses can be exclusive of each other.

We thank the reviewer for raising these important points. The antigen used was a stabilized headless H1 stem¹⁰. While the authors agree that a group 2 stem construct would have been more relevant to infection outcomes, regrettably this antigen was not available to the research team at the time these analyses were conducted. We have added this limitation to the manuscript.

14. Figure 5, can the frequency of NK cells, expression of CD57 and CD16 be shown as a function of age, as a continuous variable? Are correlations in figure 5 def, based on both younger and older NK cells, does the strength of the relationship change w age? NK cell function is also time dependent from collection (<12 hours), what time post bleed were these cells tested and were they freeze thawed?

We thank the reviewer for their attention to detail in our NK cell experiment. While we prefer the discrete graphs for legibility, we have included the continuous graphs here for review and within the supplementary information.

Figure S6. Dot plots show the percentage of NK cells positive for CD57 and expression level (MFI) of CD16 at baseline condition by age. Dots show individual subjects (<40 n=10, >65 n=19). Correlation measured by Spearman's R, *** p < 0.001.

Related to the question about time to freeze/analysis, all PBMCs from elderly individuals were cryopreserved based on a standard operating protocol established for the frailty study led by Drs. Kuchel and McElhaney. The SOP was based on decades of experience with biobanking, and processed <6 hours from collection. The cells were then thawed per Ragon Institute SOP, where Dr. Alter has extensive experience in working with NK cells, and assays were run immediately upon thaw and counting to avoid cell loss.

15. Figure 2 – Authors say that due to heterogeneity, multivariate analysis would more likely predict the differences between cases and controls, as shown here – but did authors do univariate analyses for these ADCC related binding and functions? And if so were any NK cell activities independent predictors of protection? Is this necessary for the conclusion sentence on line 1874?

We appreciate the reviewer's request, and while there was a trend towards and enrichment of ADCC-related features in controls, none of the features reached statistical significance after multiple-test correction, likely due to the small sample size. However, several features were significant prior to correction, and thus future validation studies, that solely collect NK cell features may achieve statistical significance if only these features are tested for their predictive power.

16. Table S1, this cohort information is missing age, gender, prior vaccine history data. We thank the reviewer for this clarifying question. Samples were matched for age and gender by clinical trial staff but that information was removed as part of the anonymization process prior to provision of samples to research staff. No prior vaccine history data was collected as part of this study, but no participants had received an influenza vaccine in the prior six months. This information has been added to the manuscript.

17. Line 1264 – Please include H3N2 and H1N1 cases in Table S1. Were all cases included in analysis, or just H3N2?

The authors appreciate the chance to further detail the case data. All influenza A cases identified in this study were H3N2. This has been clarified in Table S1.

Reviewer #3 (Remarks to the Author):

The manuscript entitled "Antibody-mediated NK cell activation as a correlate of immunity against influenza infection" submitted by Boudreau et al. employs systems serology to dissect markers of protective immune responses upon influenza vaccination. Comparing serum samples from vaccinees who later on got infected with influenza or did not get infected, the authors conclude that activation of NK cells plays a major role in immune protection. The study provides very valuable insights regarding Fc-mediated effector functions in protection from viral infection. I find the conclusion of the data pointing towards non-granule mediated killing as a mechanism especially interesting also related to other viral infections. Overall, the study is of high quality and the conclusions are well supported by the presented experimental data.

I, however, would kindly ask the authors to elaborate on the following points:

1. Figure 1: I am a bit confused by the different abbreviations. While the main text mentions "HA-specific Fc profiles" the figure legend refers to H3 and HAI antibodies. For a non-specialist it is unclear if that is the same. Furthermore, why were detailed results shown for H3-antibodies but not HAI antibodies? If there were no notable

correlations this could be mentioned specifically. Of note, the results section for figure 2 additionally talks about NA-antibodies which are also not mentioned in figure 1.

The authors appreciate the opportunity to add specificity to our language. HAI antibodies are specifically antibodies that mediate hemagglutination inhibition (a proxy measurement for neutralization by receptor blocking). The HA-specific Fc profiles consist of the systems serology measurements of Fc-dependent innate immune activation, antigen-specific antibody isotype levels, and antigen-specific antibody FcR binding. Specifically in Figure 1, we present H3-specific Fc profiles using the H3 HA from the vaccine strain. The NA (specifically vaccine strain N2) antibodies are also included in Figure 1C, and have been added to the text. Throughout this section of the results, we have updated language to add additional explanations of abbreviations and increase specificity.

2. Line 151-169: How do the authors imagine the do novo response to non-H3 vaccine strains to be initiated in absence of infection with the respective virus strains (subjects primarily infected with H3N2)? If these subjects have been infected with H1 virus before, I personally would not call the response "de novo". Could it be some form of bystander activation?

We thank the reviewer for the chance to further explain our reasoning. The individuals in this study received inactivated influenza vaccines including H1N1, H3N2, and B strains. Therefore, in the response to vaccination, we would expect to see antibody generation against all strains. As the reviewer correctly notes, these individuals have more than likely been infected with H1N1, H3N2, and B influenzas previously, as have most adults. We identified these responses as likely de novo responses based on the presence of IgG3 and IgM isotypes, which are produced early in the germinal center response prior to high amounts of class switching. Thus, the term "de novo" is not meant to imply that these individuals have or have not been previously exposed to a strain, but rather that the emerging humoral immune response is reflective of a newly maturing response derived from a newly formed germinal center reaction.

3. Concerning interpretation of Fc glycosylation results: The authors mention increased fucosylation and increased di-sialylation in cases over controls. In fig. 3b one can see that a prominent part of the control group show comparable fucosylation as the case group. This suggests that fucosylation might not solely account for the observed differences and I thus find the statement in lines 205-207 a bit strong. Was there a correlation of NK cell activation (and other measured parameters) with the level of fucosylation in individual samples? What about correlation of distinct glycosylation profiles within donors e.g. fucosylation and glycosylation? If possible, providing primary data of these analyses would be helpful in terms of getting further insights into the role of distinct glycosylation patterns for human IgG activity.

We thank the reviewer for their suggestions concerning our Fc glycosylation data. As the reviewer rightly identifies, there is an overlap in H3-specific IgG fucosylation observed the cases and controls. However, as the reviewer also notes, that fucosylation can occur on a large number of distinct glycan structures. As is shown in the heatmap, sialylation/di-sialylation also was strongly associated with FcGR3A binding and NK cell degranulation, as was galactosylation. Thus, more processed glycans, that are non-fucosylated are associated with more robust NK cell activation (Figure to the right). While individual structures were not resolvable in this study, future studies able to capture the spectrum of structures with higher resolution may identify a specific sub-structure that fully resolves the groups. Yet, given that antibodies bind as polyclonal swarms, it is critical to note that even small changes in glycosylation across populations may profoundly influence effector functions, in such a way that an immune complex must not be decorated exclusively with afucosylated antibodies to drive ADCC, but rather, composition of an immune complex with increasing frequencies of afucosylated antibodies may be sufficient to drive enhanced NK cell activity. This and the point related to a need for future deeper structural analysis have been added to the manuscript.

Dot plots show Spearman R correlations between measures of H3 WT-specific NK cell activation and FCGR3A binding versus H3 WT-specific antibody Fc glycoforms.

The reviewer also raises an interesting question regarding the correlation of distinct glycosylation profiles within donors. In doing these analyses, we found that fucosylation and sialylation were strongly anti-correlated, while sialylation was heavily linked to digalactosylation, as expected. These data reflect the known mechanisms of antibody Fc N-linked glycosylation¹¹. This data has been added to Figure S5.

Heat map shows Spearman correlations between different antibody Fc glycoforms. All individuals (n=100) were included in this correlation analysis.

4. With respect to differences observed in NK cell activation in vitro and more importantly protection of vaccinees I am also wondering if the authors had the opportunity to genotype for the FcγRIIIa-158V/F polymorphism as this has previously been suggested to impact IgG binding. Even if note, this could be a point to add to the discussion.

Regrettably, information on FCGR allotype polymorphism was not available in this cohort. As the reviewer correctly identifies, FCGR3A-158V has higher affinity for IgG1 and IgG3, as well as binding to IgG4 which the 158F variant does not display. We have added this as a limitation to the study.

5. In line 329, the authors propose the induction of specific NK cell functions by targeting specific receptors. This is indeed a promising perspective for future vaccines and/or therapeutics but I find the word "can" again a bit strong given our lack of knowledge on the interplay of all those receptors. I would kindly suggest to rephrase. We appreciate the reviewer's identification of this overstatement. We have adjusted this sentence to more accurately reflect our original authorial intent.

Last but not least I am curious about a couple of methodological details: We thank the reviewer for raising these important technical points.

- Why were serum samples diluted differently for ADCP and ADNP assays? Along those

lines, did the authors account for heterogeneity in antigen-specific IgG levels for serum dilutions or when analysing the data? Alternatively, were beads checked for saturation with donor IgG? If not, it might be more difficult to dissect the inflammatory potential of IgG in distinct samples independent of its concentration.

The serum samples were diluted for each assay such that samples fell within the linear range of detection. As the assay sensitivity varies by cell type, different dilution factors were required. These assay characteristics were all defined during standard operating procedure development under our GCLP laboratory. Thus, for each study/antigen, dilution factors are determined by preliminary experiments with a small subset of samples run across a wide range of dilutions. The authors agree that heterogeneity in IgG levels plays an important role in the immune response and protection from infection; which is why all samples within the cohort are run at the same dilution. However, given that titer alone does not predict protection, the data presented here suggest that functional characteristics of pathogen-specific antibodies may provide additional insights on mechanisms of potential protection against Influenza.

- Why was the ADCP assay done separately with THP-1 cell line instead of taking that data from the ADNP assay that not only included neutrophils but also monocytes upon ACK lysis of blood samples?

The reviewer asks an interesting question, as immune complexes may also be taken up by monocytes in a whole blood assay if the cells are not ruptured prior to ACK. However, over the years we have done some work to explore the possibility of examining the effect of immune complexes on various cells types in whole blood, but the results are often highly heterogeneous. Several attempts to standardize this approach to capture robust data in monocytes have failed thus far- but we hope to solve this problem in the future for clinical trial sample testing. Thus, for this study, we favored the use of a monocyte cell line, to specifically explore the role of antibodies on immune complex uptake, and examined neutrophil phagocytosis separately after ACK. We have made the point that next generation assays may continue to evolve, as recently documented for a blood monocyte-derived macrophage assay¹².

Line 26: "were monitored influenza infection" seems like a word is missing

Line 50: correlate instead of correlates?

Line 78: split sentence after "influenza"

Line 166: spelling error in specific

We thank the reviewer for identifying these typographic errors and have corrected them.

Reviewer Response References

1. DiazGranados, C. A. *et al.* Efficacy of High-Dose versus Standard-Dose Influenza Vaccine in Older Adults. *N. Engl. J. Med.* **371**, 635–645 (2014).
2. Hannoun, C., Megas, F. & Piercy, J. Immunogenicity and protective efficacy of influenza vaccination. *Virus Res.* **103**, 133–138 (2004).
3. Loeb, N. *et al.* Frailty is associated with increased hemagglutination- inhibition titers in a 4-year randomized trial comparing standard- And high-dose influenza vaccination. *Open Forum Infect. Dis.* **7**, (2020).

4. Verschoor, C. P. *et al.* Advanced biological age is associated with improved antibody responses in older high-dose influenza vaccine recipients over four consecutive seasons. *Immun. Ageing* **19**, 1–11 (2022).
5. Bournazos, S., Woof, J. M., Hart, S. P. & Dransfield, I. Functional and clinical consequences of Fc receptor polymorphic and copy number variants. *Clin. Exp. Immunol.* **157**, 244–254 (2009).
6. Bruhns, P. *et al.* Specificity and affinity of human Fc gamma receptors and their polymorphic variants for human IgG subclasses. *Receptor* **113**, 3716–3725 (2009).
7. Karsten, C. B. *et al.* A versatile high-throughput assay to characterize antibody-mediated neutrophil phagocytosis. *J. Immunol. Methods* **471**, 46–56 (2019).
8. Przemska-Kosicka, A. *et al.* Age-related changes in the natural killer cell response to seasonal influenza vaccination are not influenced by a synbiotic: A randomised controlled trial. *Front. Immunol.* **8**, 1–9 (2018).
9. Nielsen, C. M., White, M. J., Goodier, M. R. & Riley, E. M. Functional significance of CD57 expression on human NK cells and relevance to disease. *Front. Immunol.* **4**, 1–8 (2013).
10. Yassine, H. M. *et al.* Hemagglutinin-stem nanoparticles generate heterosubtypic influenza protection. *Nat. Med.* **21**, 1065–1070 (2015).
11. Jennewein, M. F. & Alter, G. The Immunoregulatory Roles of Antibody Glycosylation. *Trends Immunol.* **38**, 358–372 (2017).
12. Zohar, T. *et al.* A multifaceted high-throughput assay for probing antigen-specific antibody-mediated primary monocyte phagocytosis and downstream functions. *J. Immunol. Methods* **510**, 113328 (2022).

Reviewers' Comments:

Reviewer #1:

Remarks to the Author:

All open questions were answered sufficiently and the manuscript was adapted accordingly.

Reviewer #2:

Remarks to the Author:

The authors have done a thorough reply addressing reviewer points. However the additional mouse passive transfer experiment with CR9114 antibody with and without binding FcR function in C57BL6 mice w X31 challenge raises further technical questions. These mice do not match the Fc engagement of human FcR- humanised FcR mice developed by Bruhns/Ravtech would be more ideal. The FcR binding of mouse cells to human antibodies should be shown if their data is included, otherwise I would recommend removing the mouse experiment.

Line 422: stipulate ethnicity (95% caucasian) and FcγR3a H131 allotype as described in reply.

Reviewer #3:

Remarks to the Author:

I thank the authors for their thorough responses to the reviewer's comments and appreciate them including the suggested changes into the revised version.

I only have a couple of minor last comments:

Line 201: Please include a reference for the NK-enhanced Fc mutant.

Line 300: I fully agree especially as an impact of different adjuvants on IgG glycosylation has already been shown in mice and could be referenced here to support this statement (Bartsch et al, doi: 10.1016/j.jaci.2020.04.059; Kao et al, doi: 10.1016/j.jaci.2020.04.059)

Line 528: Do you mean HEK293F cells?

Otherwise I now fully support publication of the manuscript.

RESPONSE TO REVIEWERS' COMMENTS

Reviewer #1 (Remarks to the Author):

All open questions were answered sufficiently and the manuscript was adapted accordingly.
We thank the reviewer for taking the time to review our manuscript again.

Reviewer #2 (Remarks to the Author):

The authors have done a thorough reply addressing reviewer points. However the additional mouse passive transfer experiment with CR9114 antibody with an without binding FcR function in C57BL6 mice w X31 challenge raises further technical questions. These mice do not match the Fc engagement of human FcR- humanised FcR mice developed by Bruhns/Ravtech would be more ideal. The FcR binding of mouse cells to human antibodies should be shown if their data is included, otherwise I would recommend removing the mouse experiment.

We agree that the humanized FcR developed by Bruhns/Ravetch would be ideal for this type of experiment in the future. Regrettably, this model was not available to the authors during the study. In consultation with the editor, we have decided to remove the mouse experiment from this manuscript and save it for a future publication.

Line 422: stipulate ethnicity (95% caucasian) and FcγR3a H131 allotype as described in reply.
We thank the reviewer for the reminder to include this detail in the methods section. This has been added.

Reviewer #3 (Remarks to the Author):

I thank the authors for their thorough responses to the reviewer's comments and appreciate them including the suggested changes into the revised version.
We appreciate the reviewer's time and effort to review our paper!

I only have a couple of minor last comments:

Line 201: Please include a reference for the NK-enhanced Fc mutant.
This line has now been removed as it referenced the mouse experiment (see above).

Line 300: I fully agree especially as an impact of different adjuvants on IgG glycosylation has already been shown in mice and could be referenced here to support this statement (Bartsch et al, doi: 10.1016/j.jaci.2020.04.059; Kao et al, doi: 10.1016/j.jaci.2020.04.059)
We thank the reviewer for drawing our attention to these two excellent publications. The references have been added.

Line 528: Do you mean HEK293F cells?
We thank the reviewer for catching this typo – it has been corrected.

Otherwise I now fully support publication of the manuscript.